# Swinging motion of a kite with suspended control unit flying turning manoeuvres

Mark Schelbergen[1] and Roland Schmehl[1]

[1]Faculty of Aerospace Engineering, Delft University of Technology, 2629 HS Delft, Netherlands

**Correspondence:** Mark Schelbergen (m.schelbergen@tudelft.nl)

**Abstract.** The flexible membrane kite employed by some airborne wind energy systems uses a suspended control unit, which experiences a characteristic swinging motion relative to the top of the kite during sharp turning manoeuvres. This paper assesses the accuracy of a two-point kite model in resolving this swinging motion using two different approaches: approximating the motion as a transition through steady-rotation states and solving the motion dynamically. The kite is modelled with two rigidly linked point masses representing the control unit and wing, which conveniently extend a discretised tether model. The tether-kite motion is solved by prescribing the trajectory of the wing point mass to replicate a figure-of-eight manoeuvre from the flight data of an existing prototype. The computed pitch and roll of the kite are compared against the attitude measurements of two sensors mounted to the wing. The two approaches compute similar pitch and roll angles during the straight sections of the figure-of-eight manoeuvre and match measurements within three degrees. However, during the turns, the dynamically solved pitch and roll angles show systematic differences compared to the steady-rotation solution. As a two-point kite model resolves the roll, the lift force may tilt along with the kite, which is identified as the driving mechanism for turning flexible kites. Moreover, the two-point kite model complements the aerodynamic model as it allows computing the angle of attack of the wing by resolving the pitch. These characteristics improve the generalisation of the kite model compared to a single-point model with little additional computational effort.

## 1 Introduction

Pumping airborne wind energy (AWE) systems with flexible membrane kites are reaching a technology readiness level suitable for first commercial applications. Two prominent examples are the leading developers SkySails Power GmbH using ram-air kites and Kitepower B.V. using leading-edge inflatable (LEI) kites (Vermillion et al., 2021; Fagiano et al., 2022). Both systems employ a single tether and a suspended kite control unit (KCU) for actuating the wing, as illustrated in Fig. 1. At the present stage of development, AWE systems are not optimised yet in terms of power production. Instead, the priority is improving operational reliability and demonstrating long-term operation, as well as learning how the systems perform in different wind environments (Salma et al., 2019). This knowledge will be crucial for designing the next generation of systems with increased power output.

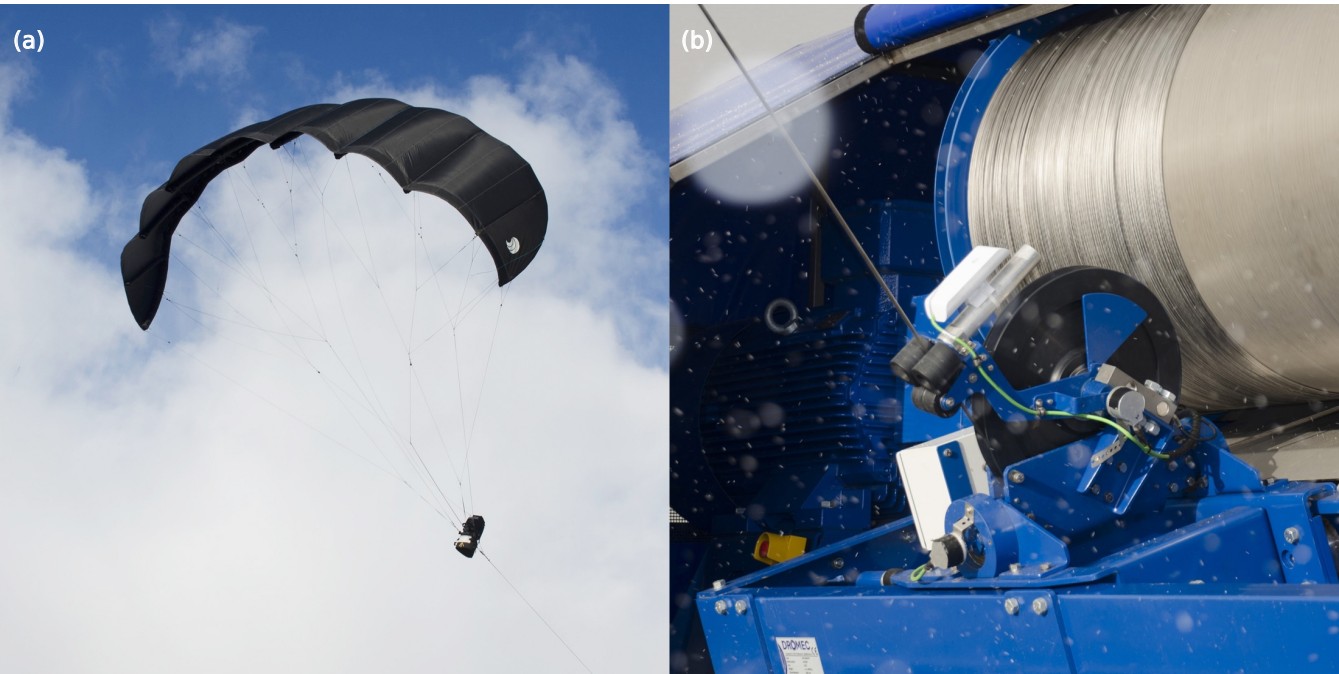

**Figure 1.** AWE system with (**a**) 25 m$^2$ V3.25B kite and (**b**) 100 kW ground station in operation in 2018 (photos courtesy of Kitepower B.V.).

Performance models estimate the energy generation of a specific system in a varying wind environment representative of the wind climate at a specific site. The models can be classified according to how the flight trajectory and the system configuration are represented. The modelled system configuration includes at least the kite and the tether.

The most simple flight trajectory representations idealise the flight with only one or a couple of steady flight states calculated with Loyd's analytic theory of tethered flight (Loyd, 1980) or derivatives thereof. The more refined quasi-steady models prescribe a parameterised flight path. Thereby, they only account for the effect of turning but do not necessarily describe the turning mechanism. Alternatively, dynamic models solve the flight path, which requires incorporating a turning mechanism. Dynamics models can be applied in an optimal control problem (OCP) to find an optimal realistic flight path.

The simpler system model configurations represent the kite as a single point mass or rigid body and assume a straight tether with its mass and drag lumped to the kite point mass. More refined models also resolve tether sag induced by lateral forces on the tether, such as gravity, centrifugal force, and aerodynamic drag, often achieved through discretising the tether. Typically, the discretised tether is represented with lumped masses connected with rigid links or spring-damper elements (Gohl and Luchsinger, 2013; Fechner et al., 2015; Rapp et al., 2019; Williams et al., 2007; Zanon et al., 2013). Alternatively, Sánchez-Arriaga et al. (2019) apply a multi-body approach using rigid rods. Fechner et al. (2015) expand the discretisation approach to the kite. The kite is represented with five point masses; four point masses represent the wing, and one additional point mass represents the suspended KCU. A lumped-mass model with spring-damper elements is considered too computationally costly for performance calculation but very useful for control system design.

Williams (2017) employs separate solvers for computing the motion of the kite and the tether. The 'quasi-static' tether deformation is solved as a subroutine to solving the flight trajectory to efficiently account for tether elasticity. Consequently, the tether deformation due to gravity, centrifugal force, and aerodynamic drag is considered, while transient motion and longitudinal vibration are neglected. The discretised tether model assumes that the entire airborne system, including tether and kite, jointly rotates around the tether attachment point at the ground. This assumed kinematic relationship works well for near-straight flights but is not representative during turning manoeuvres.

The choice of the kite model determines the level of abstraction required to introduce steering forces as demonstrated in the work of Fechner et al. (2015). The work presents both a single-point and a five-point kite model of a LEI kite. By resolving the roll of the kite with respect to the upper tether element, the five-point kite model allows for realistically incorporating the centripetal force acting on the relatively heavy control unit. Additionally, the outboard-pointing lift forces produced by the wing tips contribute to the centripetal force. Contrastingly, the single-point model requires making large assumptions about the composition of the aerodynamic forces due to the lack of information about the attitude of the kite. The definition used for the lift force generated by the top wing surface does not enable the lift force to contribute to the centripetal force. To enable steering, the single-point model employs an artificial side force proportional to the steering input, which necessitates a disproportionately high side force coefficient.

In reality, the aerodynamics of the kite highly depend on the fluid-structure interaction involving the membrane wing and bridle. The LEI kite of Kitepower B.V. is steered by pulling the rear bridle lines attached to one side of the wing while loosening the lines on the other side. This asymmetric actuation of the bridle line system makes the wing deform and initiate a turn. Video footage of experiments shed some light on the aero-structural deformation due to steering (Schmehl and Oehler, 2018). Previous research on the topic has focused mainly on the interaction between the flow and the deforming membrane wing (Breukels et al., 2013; Bosch et al., 2013; Geschiere, 2014; Duport, 2018; Oehler et al., 2018; Thedens, 2022; Folkersma, 2022; Poland and Schmehl, 2023; Cayon et al., 2023). The experimental data also indicates a pronounced dynamic interaction between the wing, the suspended KCU, and the tether during the turning manoeuvre. How much the swinging motion of the KCU relative to the wing affects the turning behaviour and the power generation of the kite has only recently been studied by Roullier (2020). An improved understanding of this effect would allow for enhancing performance models of flexible membrane kites, designing more precise control algorithms, and ultimately improving the system performance.

The goal of this paper is twofold: to study the dynamics that induce the observed characteristic pitch and roll swinging motion of the kite during sharp turning manoeuvres and discuss the implications to performance modelling. Pertaining to the first goal, this paper introduces a two-point kite model that is used together with a straight and discretised tether. Firstly, the motion is approximated as a transition through steady-rotation states with both tether representations. Subsequently, the motion is resolved dynamically with the discretised tether to study the impact of transient effects. Instead of resolving the translational motion of the wing, we prescribe a cross-wind flight path from the flight data of Kitepower B.V. This removes the dependency of the model on the aerodynamics of the kite and, thereby, reduces uncertainties. Pertaining to the second goal, this paper provides a breakdown of the mechanisms that initiate and drive a turn of a flexible kite system with a suspended control unit.

This paper is organised as follows. In Sec.2, the experimental data underlying this study is described. In Sec.3, the computational models are outlined. The results are presented in Sec. 4 and discussed in Sec.5. Conclusions are presented in Sec. 6.

## 2    Test flight data

The data used in the present study was acquired on 8 October 2019 using a $25\,\text{m}^2$ V3.25B kite of Kitepower, depicted in Figs. 1 and 2. This derivative of the TU Delft LEI V3 kite was already investigated by Oehler and Schmehl (2019) and is

illustrated in Fig. 3. We use the term kite for the entire assembly of the wing, bridle line system, and suspended control unit. The V3.25B kite is substantially smaller and less performant than the $60\,\text{m}^2$ kite of Kitepower's commercial 100 kW system that is being developed at the time of writing. Moreover, conservative operational settings were used for this specific flight because its purpose was to test new hardware and software components of the system and to acquire data. Consequently, the power output during the test was substantially lower than the nominal power output of the commercial system.

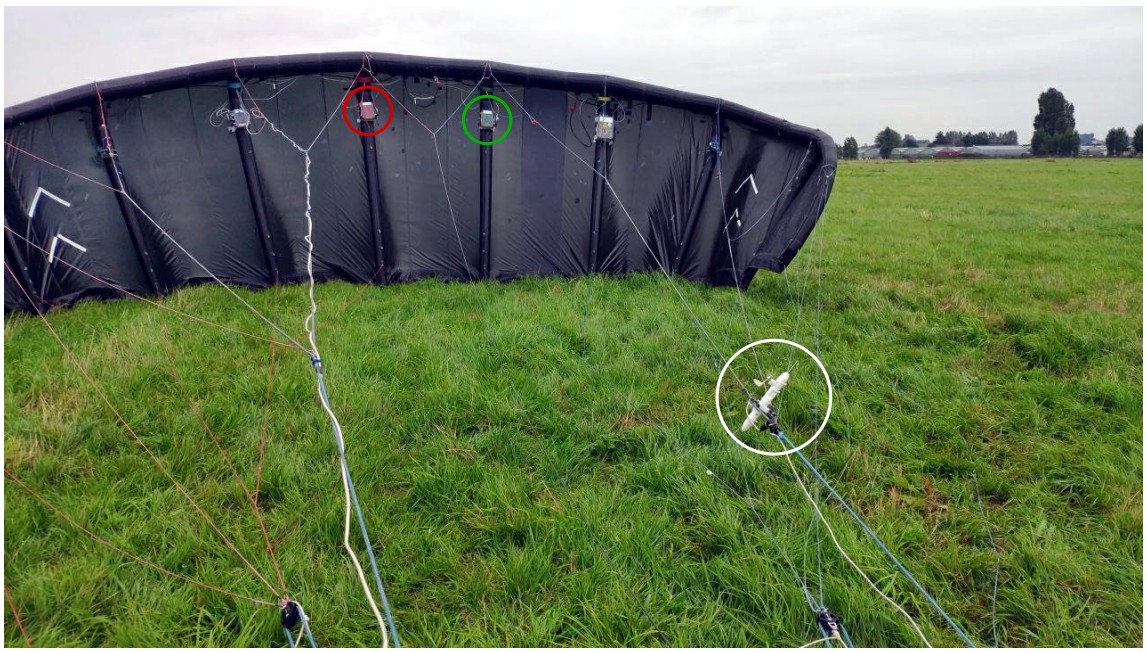

**Figure 2.** Fully instrumented V3.25B kite before launch (photo courtesy of Kitepower B.V.). The overlaid red, green, and white circles mark Pixhawk® sensor 0, Pixhawk® sensor 1, and the flow sensors, respectively.

The published data set (Schelbergen et al., 2024) covers approximately three hours of flight time, during which 87 automatic pumping cycles were recorded. With this comprehensive collection of data, statistical insights into the flight behaviour of the kite can be gained. The apparent wind speed was measured with a Pitot tube attached to the front bridle lines at the connection to a power line. This flow sensor is visible in the foreground of Fig. 2, also featuring a flow vane to measure the angle of attack. The side slip angle was not measured in this setup. The onboard electrical power was supplied by a small ram-air turbine

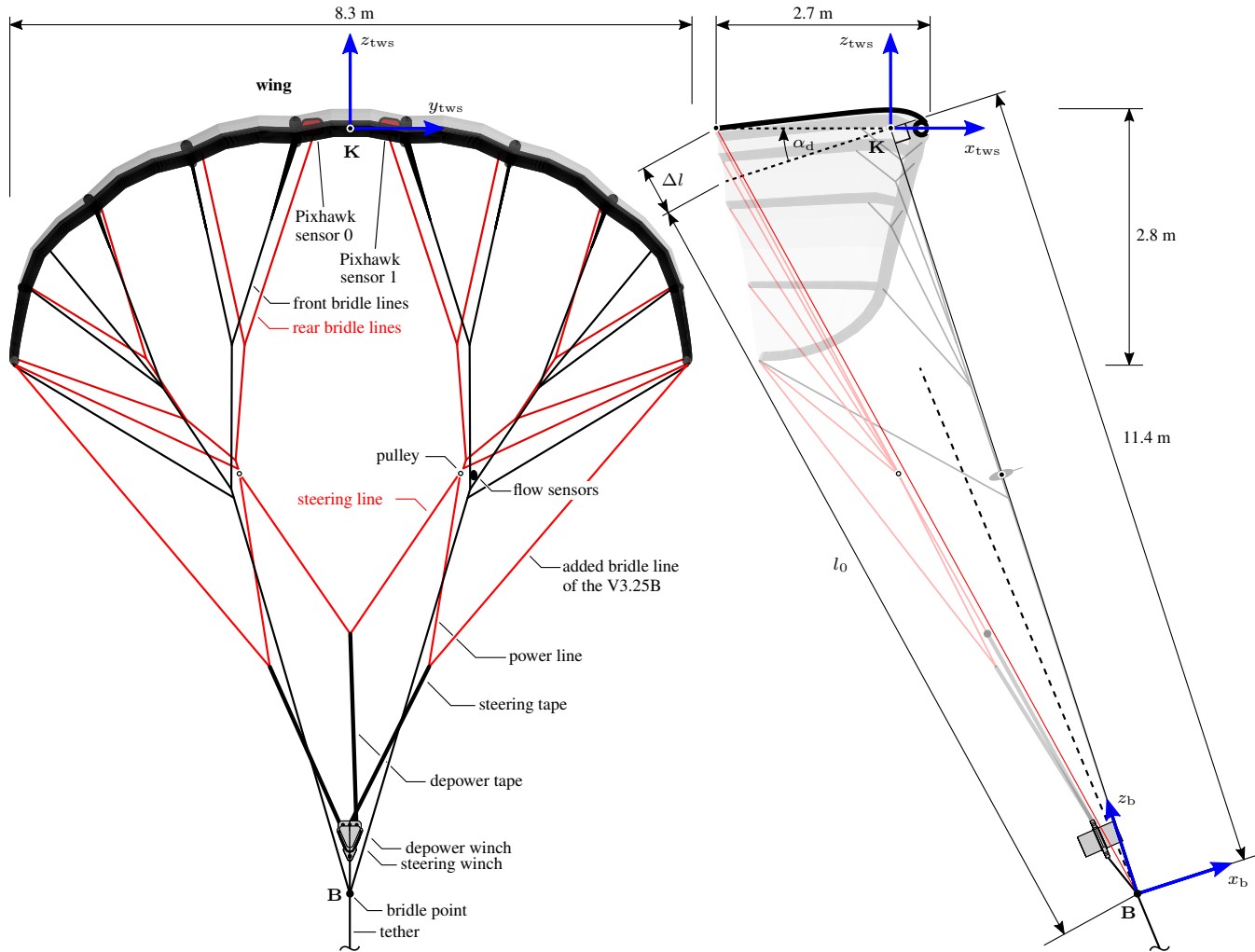

**Figure 3.** Front-view (left) and side-view (right) of the CAD geometry of the V3.25B kite in depowered state. Also depicted are the top wing surface (TWS) reference frame $x_{\mathrm{tws}}, y_{\mathrm{tws}}, z_{\mathrm{tws}}$, with origin $\mathbf{K}$ at the point around which the wing pitches when changing the angle of attack, and the bridle reference frame $x_{\mathrm{b}}, y_{\mathrm{b}}, z_{\mathrm{b}}$ with origin at the bridle point $\mathbf{B}$. The positions of the two Pixhawk® sensors 0 and 1 are assumed to be fixed with respect to the TWS reference frame while the relative flow sensors are assumed to be fixed with respect to the bridle reference frame. Adapted from Oehler and Schmehl (2019).

mounted to the KCU, as shown in Peschel et al. (2017). A tether with a diameter of 10 mm was used for the flight test. The tether force and the reel-out speed were measured at the ground station.

     For this flight test, two Pixhawk® sensor units were mounted to the wing, one on each of the two struts adjacent to the symmetry plane of the kite (red and green cases in Fig. 2). The units are each equipped with an IMU, GPS sensor, and barometer for recording position and attitude. Figure 4a–d depict the conditioned position data of one figure-of-eight cross-

wind manoeuvre from the flight data made available by Kitepower. The position data is based on measurements of sensor 0, which have been processed using the default Kalman filter implementation of the Pixhawk®. The velocity measurements used in the present analysis come from the same sensor. The tangential and radial components of these measurements are depicted together with those measured by sensor 1 in Fig. 4e–f (decomposition shown in Fig. 6). For an unknown reason, sensor 0 did not measure acceleration. Therefore, the acceleration measured with sensor 1 is used in the analysis and is depicted in Fig. 4g–i.

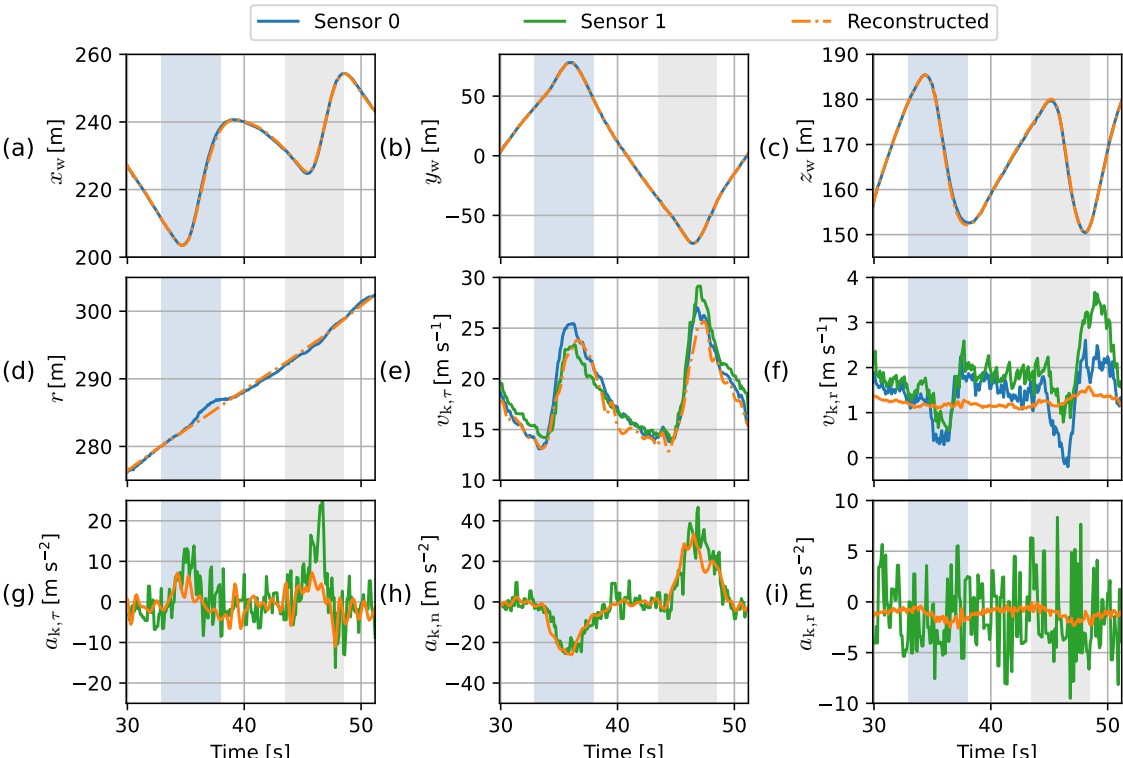

**Figure 4.** Kinematics of the studied figure-of-eight manoeuvre measured with the two Pixhawk® sensor units and the kinematics obtained with the flight trajectory reconstruction described in Appendix A. The intervals shaded blue and grey indicate left and right turns, respectively. (**a–c**) Kite position coordinates of the wind reference frame (sensor data is Kalman filtered). (**d**) Radial position coordinate of the kite. (**e–f**) Tangential and radial kite velocity. (**g–i**) Tangential, normal, and radial kite acceleration. The unit vectors defining the directions used for the decomposition are depicted in Fig. 6.

Comparing the tether reel-out speed to the position of the wing indicates anomalies in the recorded wing position that manifest as unrealistically large jumps in radial position predominately occurring during left turns, as can be observed in Fig. 4d. These anomalies are removed using a discrete-time optimisation problem that minimises the error between the modelled radial wing speed and recorded tether reel-out speed while limiting the bias between the modelled and recorded wing position. The flight trajectory reconstruction might not be strictly valid. Nevertheless, it serves the higher aim of this study by providing a

consistent kinematic input for the dynamic simulation. The identification of these anomalies and the details of the optimisation are described in Appendix A.

We illustrate our analysis using a figure-of-eight cross-wind manoeuvre of the wing shown in Fig. 5. This specific manoeuvre is part of the $65^{\text{th}}$ pumping cycle of the test flight. Because of the high repeatability of the automatic flight manoeuvres, the other figures of eight of the dataset give similar results. Characteristic reference positions along this manoeuvre are designated

to highlight the analysis, listed in Table 1. The kite flies along the trajectory in the direction of increasing reference numbers, i.e., flying upwards on the straight path segments and downwards during the turns. The tether is reeled out while the kite is flying cross-wind manoeuvres, increasing the radial position of the kite from 276 to 302 m at a height of 150–185 m. The asymmetry of the trajectory may be explained by various factors, such as the misalignment with the wind velocity due to wind veer and imperfections within the system.

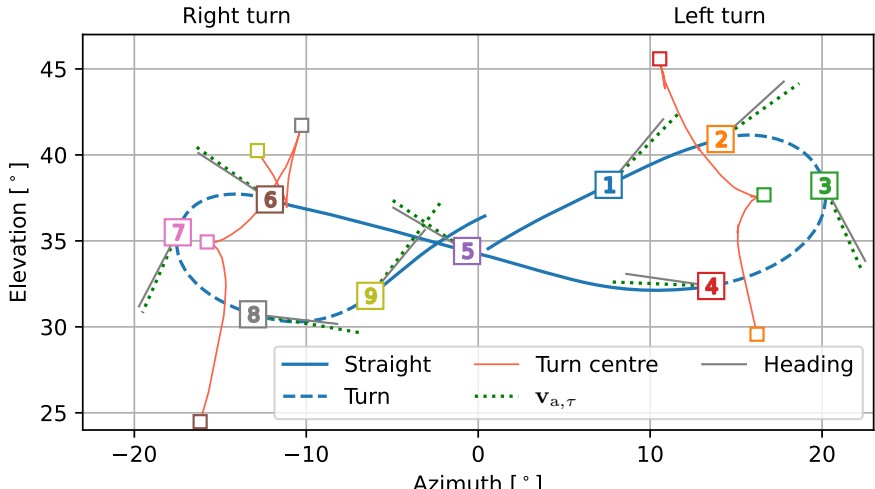

**Figure 5.** The studied figure-of-eight cross-wind manoeuvre of the wing depicted with respect to the wind reference frame, shown in Fig. B1. The flight path is composed of straight (solid blue) and turn (dashed blue) line segments. Reference positions 1 to 9 are designated along the path in flight direction. For the two turns, the changing position of the turn centre is tracked with the red lines. The turn-centre markers pair with the numbered path markers of the same colour. The dotted lines depict the modelled tangential apparent wind velocity. Alongside the apparent wind velocity lines, the solid lines depict the heading inferred from the attitude measurements of sensor 1. Note that the turn at a positive azimuth is observed as a left turn looking from the ground station to the kite, while the turn at a negative azimuth is observed as a right turn.

For simplicity, the present study assumes that the wind velocity is uniform and constant. The average wind speed measured at the ground for the reference pumping cycle is approximately $7 \text{ m s}^{-1}$. Based on the estimated wind shear, the wind speed at the kite is assumed to be $10 \text{ m s}^{-1}$. The grey lines in Fig. 5 show the heading of the kite at the reference positions inferred from sensor 1. The dotted green lines show the projection of the approximated apparent wind velocity computed as

$$\mathbf{v}_a = \mathbf{v}_w - \mathbf{v}_k, \tag{1}$$

**Table 1.** Timestamps of the reference positions along the figure-of-eight path shown in Fig. 5, starting at 29.9 s and ending at 51.2 s in the $65^{\text{th}}$ pumping cycle.

| Instance label | 1 | 2 | 3 | 4 | 5 | 6 | 7 | 8 | 9 |
|---|---|---|---|---|---|---|---|---|---|
| Time [s] | 31.9 | 33.9 | 35.6 | 37.5 | 41.0 | 44.5 | 46.2 | 47.6 | 49.1 |

in which $\mathbf{v}_{\text{w}} = [10\,0\,0]^{\top}$ is the wind velocity in the wind reference frame and $\mathbf{v}_{\text{k}}$ is the measured kite velocity. The side slip angle is the angle between the heading of the kite and the apparent wind velocity. The approximation of the apparent wind velocity lacks the necessary precision to assess the side slip. Moreover, the side slip angle was not measured during the flight test, and assessing the side slip is out of scope.

## 3 Computational modelling

The flight behaviour along the figure of eight described in the previous section is analysed with two different methods for solving the motion of the two-point kite model with a discretised tether model. First, this section discusses the tether-kite model configuration. Next, the two methods for solving the motion are discussed. The first approximates the tether-kite motion as a transition through steady-rotation states. The second solves the motion dynamically.

### 3.1 Tether-kite model

The two-point kite model accounts for the two distinct mass concentrations of the wing and the KCU. During cross-wind flight, the bridle line system is tensioned by the aerodynamic force acting on the wing. Accordingly, the two point masses stay at a constant distance, considering that the effect of wing actuation, including deformation, is negligible. From a modelling perspective, the two rigidly-linked point masses resemble a rigid body model, with rotational inertia in pitch and roll but not in yaw. The yaw motion is irrelevant to the present analysis due to the exclusion of the wing aerodynamics. This would not be the case when solving the full, unconstrained kite motion.

The two-point kite model developed for the present analysis can be added in a straightforward way to a discretised tether model as an additional final element. An example with five tether elements of equal length $l_j$ and a kite element of length $l_{\text{b}}$ is shown in Fig. 6. To account for a varying length $l_{\text{t}}$ and mass $m_{\text{t}}$ of the deployed tether, the element lengths and point masses are updated every instance according to

$$l_j = \frac{l_{\text{t}}}{N}, \tag{2}$$

$$m_j = \frac{m_{\text{t}}}{N}, \tag{3}$$

where $N$ is the constant number of tether elements. The point mass representing the KCU is determined as

$$m'_{\text{kcu}} = m_{\text{kcu}} + \frac{m_j}{2}. \tag{4}$$

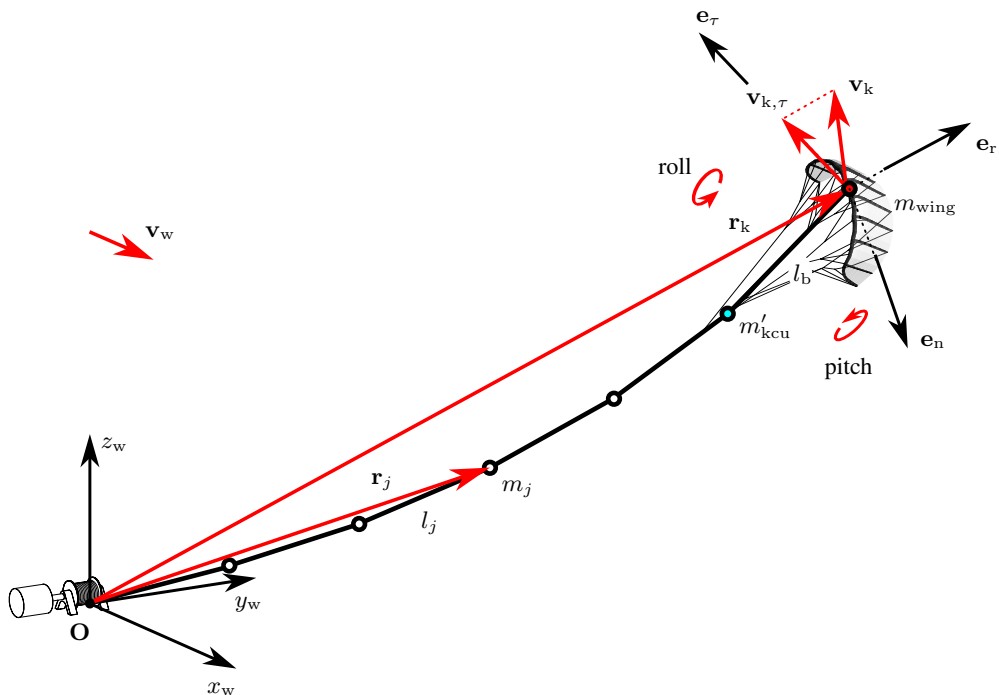

**Figure 6.** Two-point model of the kite expanding a tether discretised by $N = 5$ tether elements. The kite position $\mathbf{r}_k$ and flight velocity $\mathbf{v}_k$ are defined at the point $\mathbf{K}$ where the sensor units are attached to the wing, see Fig. 3. Also shown are the tangential kite velocity component $\mathbf{v}_{k,\tau}$, the unit vectors $\mathbf{e}_\tau, \mathbf{e}_n, \mathbf{e}_r$ used to decompose the measured kite velocity and acceleration, and the wind reference frame $x_w, y_w, z_w$ with origin at the tether attachment point $\mathbf{O}$ on the ground and $x_w$-axis aligned with the wind velocity vector.

The tether and kite elements are assumed to be rigid. Moreover, variations in the lengths of these elements due to elasticity are neglected. The effect of tether elasticity on the swinging motion of the kite is expected to be negligible as long as the modelled tether length agrees with the effective real-world tether length.

Aerodynamic drag is one of the forces considered to act on the point masses representing the tether. The drag is calculated as

$$\mathbf{D}_{t,j} = \frac{1}{2} \rho \|\mathbf{v}_{a\perp,j}\| \mathbf{v}_{a\perp,j} C_{D,t} l_j d_t, \qquad \text{with} \quad j = 1, \dots, N, \tag{5}$$

where $\rho$ is the air density, $\mathbf{v}_{a\perp,j}$ is the local apparent wind velocity perpendicular the tether element below the $j^{\text{th}}$ point mass, $C_{D,t}$ is the tether drag coefficient, and $d_t$ is the tether diameter.

Two aerodynamic forces are acting on the KCU point mass below the wing: the drag of the KCU itself $\mathbf{D}_{kcu}$ and half the drag of the upper tether element. Consequently, the total drag acting on the KCU point mass is

$$\mathbf{D}'_{kcu} = \mathbf{D}_{kcu} + \mathbf{D}_{t,kcu} = \frac{1}{2} \rho \|\mathbf{v}_{a\perp,kcu}\| \mathbf{v}_{a\perp,kcu} C_{D,kcu} A_{kcu} + \frac{\mathbf{D}_{t,N}}{2}, \tag{6}$$

in which $\mathbf{v}_{\mathrm{a}\perp,\mathrm{kcu}}$ is the perpendicular component of the apparent wind velocity at the KCU. The frontal area of the KCU is denoted as $A_{\mathrm{kcu}}$ and the drag coefficient as $C_{\mathrm{D,kcu}}$. The chosen value of 1.0 for the drag coefficient is within the typical range for a blunt body. The drag from the bridle and ram-air turbine are not incorporated explicitly, but their influence is indirectly accounted for through the KCU and wing drag. The values of physical parameters are listed in Table 2.

**Table 2.** Physical parameters of the airborne system model.

| $v_{\mathrm{w}}$ | $m_{\mathrm{kcu}}$ | $m_{\mathrm{wing}}$ | $l_{\mathrm{b}}$ | $\rho$ | $d_{\mathrm{t}}$ | $C_{\mathrm{D,t}}$ | $A_{\mathrm{kcu}}$ | $C_{\mathrm{D,kcu}}$ |
|---|---|---|---|---|---|---|---|---|
| $10\ \mathrm{m\,s^{-1}}$ | 25 kg | 14.2 kg | 11.5 m | $1.225\ \mathrm{kg\,m^{-3}}$ | 10 mm | 1.1 | $0.25\ \mathrm{m^2}$ | 1.0 |

Equation (5) does not account for any variation of the apparent wind velocity along the tether element and is only a reasonable approximation when using many tether elements. For single-element use, the alternative expression for the tether drag contribution (last term in Eq. (6)) better preserves the moment of the tether drag around the ground station

$$\mathbf{D}_{\mathrm{t,kcu}} = \frac{1}{8}\rho\,\|\mathbf{v}_{\mathrm{a}\perp,\mathrm{kcu}}\|\,\mathbf{v}_{\mathrm{a}\perp,\mathrm{kcu}}\,C_{\mathrm{D,t}}\,l_{\mathrm{t}}\,d_{\mathrm{t}}\,. \tag{7}$$

## 3.2 Steady-rotation state

The subroutine for solving the 'quasi-static' tether shape proposed by Williams (2017) is adopted in the present analysis to assess the swinging motion of the kite. With an initial guess of the tether length and orientation of the lower element, the corresponding tether shape is determined using a shooting method. The positions of the point masses are determined one by one, starting with the lowest point mass and moving up towards the last point mass located at the tether end. From the pseudo force balance on a particular point mass (at the intersection of two tether elements), the position of the next point mass is inferred. This balance considers the tensile forces, drag, weight, and centrifugal force. Given the tensile force acting on the tether element below the point mass, only the tensile force acting on the tether element above remains unknown and is solved. The direction of this force dictates the axial direction of the corresponding tether element. Together with the length of a tether element, the axial direction yields the position of the next point mass. By repeating this calculation for each point mass, the position of the kite is obtained given the measured tether force at the ground. A least squares optimisation is employed to find the tether length and shape for which the upper tether end coincides with the position of the wing. Consult Williams (2017) for more details.

To facilitate the calculation of loads, the velocities and accelerations of the point masses are approximated by assuming that they collectively rotate around the tether attachment point at the ground with a constant angular velocity $\boldsymbol{\omega}$, treating the point masses as particles lying on a rigid body. According to this kinematic assumption, the velocity and centripetal acceleration of each point mass depend solely on the angular velocity and its respective position. The velocity $\mathbf{v}_j$ and acceleration $\mathbf{a}_j$ of the $j^{\mathrm{th}}$ point mass are

$$\mathbf{v}_j = \boldsymbol{\omega} \times \mathbf{r}_j\,, \qquad \text{with} \quad j = 1,\ldots,N \tag{8}$$

$$\mathbf{a}_j = \boldsymbol{\omega} \times \mathbf{v}_j\,, \qquad \text{with} \quad j = 1,\ldots,N \tag{9}$$

where $\mathbf{r}_j$ is the position of the point mass. This kinematic assumption is referred to as the steady-rotation assumption throughout this paper.

Prior to calculating the kinematics of the point masses, the angular velocity needs to be determined. Williams approximates the rotational velocity with

$$\boldsymbol{\omega}_{\text{straight}} = \frac{\mathbf{r}_k \times \mathbf{v}_k}{\|\mathbf{r}_k\|^2} = \frac{\mathbf{r}_k \times \mathbf{v}_{k,\tau}}{\|\mathbf{r}_k\|^2}, \tag{10}$$

in which $\mathbf{r}_k$ and $\mathbf{v}_k$ are the position and velocity of the kite, respectively, and $\mathbf{v}_{k,\tau}$ is the tangential component of the kite velocity, shown in Fig. 6. The resulting rotational velocity yields a rotation along a great circle on the surface of a sphere, as shown in Fig. 7. This rotational velocity is labelled as 'straight' because the great-circle rotation produces the straight path segments of a figure-of-eight manoeuvre. Note that this rotational velocity is perpendicular to the position and the (tangential) velocity of the kite, which we refer to as the normal direction.

A shortcoming of this great-circle angular velocity approximation is that it does not yield an acceleration representative of a turning kite. Calculating the corresponding acceleration according to the steady-rotation assumption (Eqs. (8) and (9)) will yield an acceleration that is aligned with the position vector and, thus, no lateral acceleration. The lateral acceleration, however, is important to consider as it is the dominant component during turns, as can be observed in Fig. 4h. The kinematic assumption does allow a lateral acceleration; however, this requires that the angular velocity has a radial component.

The addition of a radial component to the great-circle angular velocity approximation enables producing a rotation along a small circle on the surface of a sphere coinciding with the turn of the figure-of-eight manoeuvre as shown in Fig. 7. Similar to the derivation of the normal angular velocity from Eq. (8), the radial angular velocity is derived from Eq. (9) and can be calculated with the normal component of the acceleration $\mathbf{a}_{k,n}$

$$\boldsymbol{\omega}_{\text{r}} = \frac{\mathbf{v}_{k,\tau} \times \mathbf{a}_{k,n}}{\|\mathbf{v}_{k,\tau}\|^2}. \tag{11}$$

The newly proposed rotational velocity approximation for turns reads as

$$\boldsymbol{\omega}_{\text{turn}} = \boldsymbol{\omega}_{\text{straight}} + \boldsymbol{\omega}_{\text{r}}. \tag{12}$$

The wing kinematics resulting from the flight path reconstruction are used to calculate the rotational velocity for turns. Figure 8a shows that the normal component of the turn rotational velocity is much smaller than the radial component. Figure 8b-c show the kinematics back-calculated with the steady-rotation assumption. The back-calculated wing velocity is solely produced by the normal component of the turn rotational velocity and only has a tangential component. Although the original wing velocity does have a radial component (smaller than $1.6 \text{ m s}^{-1}$) and the back-calculated speed does not, their magnitudes are virtually the same. The back-calculated wing acceleration is solely produced by the large radial component of the turn rotational velocity. The back-calculated wing acceleration also shows a very good match with the original wing acceleration despite the fact that it does not have a tangential component. In conclusion, these results show that the steady-rotation assumption with the newly proposed rotational velocity approximation yields a very good approximation of the kite kinematics.

To conclude, we incorporate the following model modifications with respect to the model of Williams (2017):

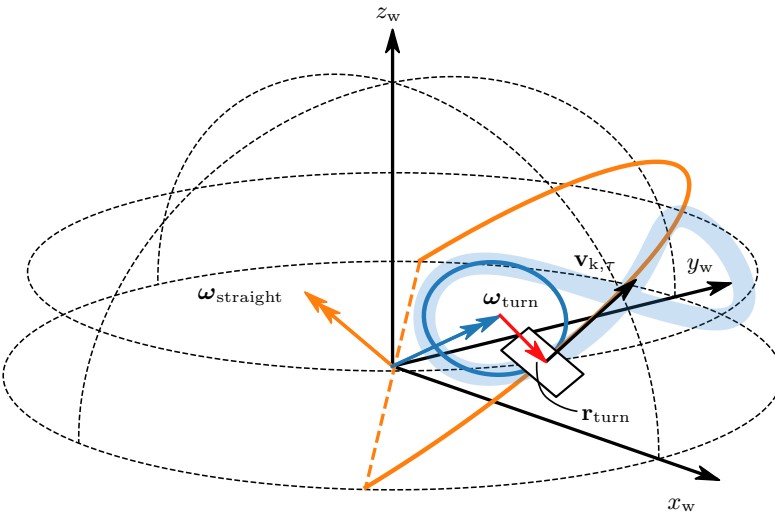

**Figure 7.** Two possible angular velocities, $\boldsymbol{\omega}_{\text{straight}}$ and $\boldsymbol{\omega}_{\text{turn}}$, that can be deduced from the tangential kite velocity $\mathbf{v}_{\text{k},\tau}$. Their respective steady-rotation flight paths comprise a great circle (orange) and an instantaneous turn circle (blue) that approximately coincides with the turn of the figure-of-eight manoeuvre. The yawed tangential plane perpendicular to the position vector of the kite is depicted as a rectangle and represents the kite.

– The elasticity of the tether elements is not considered;

   – We add a radial component to the great-circle angular velocity;

   – A different lumping approach is used for the uppermost tether point mass than for the other tether point masses, i.e., the mass and drag of half a tether element are allocated to the former instead of the mass and drag of a full element;

   – We add an extra element (rigid link) to represent the kite as described in Sec. 3.1.

**3.3   Dynamic equations of motion**

The proposed dynamic model is a derivative of the generic model for multiple kite system architectures with fixed tether lengths introduced by Zanon et al. (2013). This model uses Cartesian coordinates to reduce the non-linearity of the model formulation. Although the model allows for complex systems, we only consider a simple single-tether, single-kite configuration. The system of equations used for generating results is based on a two-point kite model and a 30-element tether model. For brevity, here
we only write out the system of equations of a model with two tether elements. The first tether element connects the ground station to the only designated tether point mass $m_1$, and the second tether element connects $m_1$ to the point mass of the control unit $m'_{\text{kcu}}$, in a similar arrangement as the configuration depicted in Fig. 6.

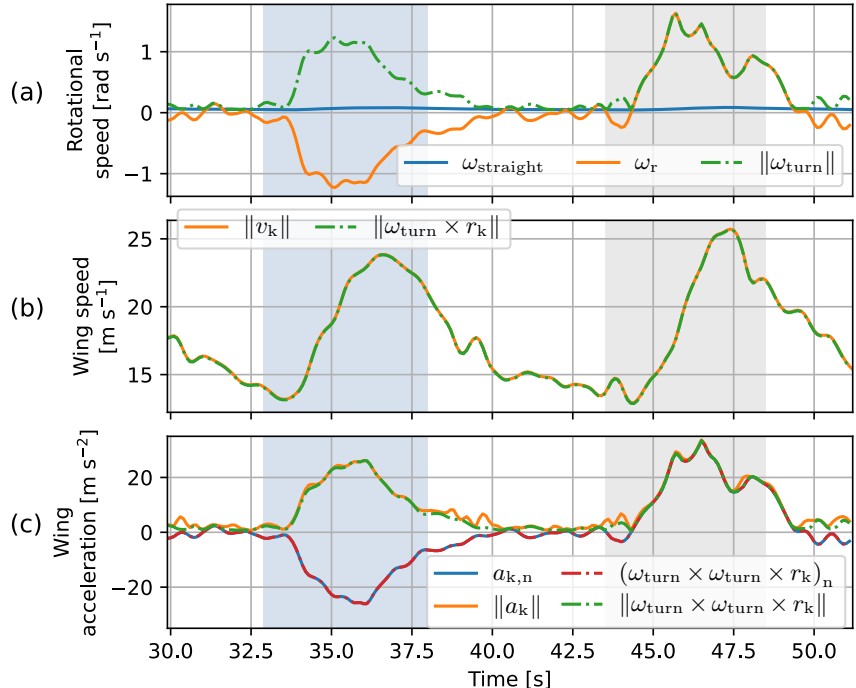

**Figure 8.** Assessing the steady-rotation assumption with the rotational velocity for turns. (**a**) The normal (straight-path) and radial rotational speed inferred from the reconstructed wing kinematics. (**b**, **c**) The wing speed and acceleration back-calculated with the turn rotational velocity (using Eqs. (8) and (9)) compared to the wing speed and acceleration from the flight trajectory reconstruction used to calculate the rotational velocity. The shaded intervals indicate the turns.

The model is described by a differential-algebraic system of equations (DAE), with constraints originating from the use of non-minimal coordinates. The differential states $\mathbf{x}$, algebraic states $\mathbf{z}$, and control inputs $\mathbf{u}$ of the two-point model are

$$\mathbf{x} = [\mathbf{r}_1,\, \mathbf{r}_{\mathrm{kcu}},\, \mathbf{r}_{\mathrm{k}},\, \mathbf{v}_1,\, \mathbf{v}_{\mathrm{kcu}},\, \mathbf{v}_{\mathrm{k}},\, l_{\mathrm{t}},\, \dot{l}_{\mathrm{t}}], \quad \mathbf{z} = [\mathbf{a}_1,\, \mathbf{a}_{\mathrm{kcu}},\, \lambda_1,\, \lambda_2,\, \lambda_{\mathrm{b}}], \quad \text{and} \quad \mathbf{u} = [\mathbf{a}_{\mathrm{k}},\, \ddot{l}_{\mathrm{t}}]; \tag{13}$$

in which subscript kcu refers to the kite control unit, k refers to the top wing surface of the kite, t denotes tether, b denotes bridle, and the numbers refer to the tether point masses and elements. The state variables are the positions $\mathbf{r}$ and velocities $\mathbf{v}$ of the point masses and the tether length $l_{\mathrm{t}}$ and reel-out speed $\dot{l}_{\mathrm{t}}$. The algebraic variables include the acceleration of the control unit point mass $\mathbf{a}$ and Lagrange multipliers $\lambda$. The Lagrange multipliers enforce the constraints and have a close relationship with the forces acting in the tether and kite elements. The control variables are the wing acceleration $\mathbf{a}_{\mathrm{k}}$ and the reel-out acceleration of the tether $\ddot{l}_{\mathrm{t}}$.

Without imposing the translational motion of the wing, the dynamics of the two-point kite model with two tether elements read as

$$
\left[\begin{array}{cccc}
\begin{bmatrix} m_1 \mathbf{I}_3 & \mathbf{0}_{3\times3} & \mathbf{0}_{3\times3} \\ \mathbf{0}_{3\times3} & m'_{\mathrm{kcu}} \mathbf{I}_3 & \mathbf{0}_{3\times3} \\ \mathbf{0}_{3\times3} & \mathbf{0}_{3\times3} & m_{\mathrm{k}} \mathbf{I}_3 \end{bmatrix} & \mathbf{G}_X^\top \\
\mathbf{G}_X & \mathbf{0}_{3\times3}
\end{array}\right]
\begin{bmatrix} \mathbf{a}_1 \\ \mathbf{a}_{\mathrm{kcu}} \\ \mathbf{a}_{\mathrm{k}} \\ \lambda_1 \\ \lambda_2 \\ \lambda_{\mathrm{b}} \end{bmatrix}
=
\begin{bmatrix}
\mathbf{D}_{\mathrm{t},1} - m_1\, g\, \mathbf{e}_z \\
\mathbf{D}'_{\mathrm{kcu}} - m'_{\mathrm{kcu}}\, g\, \mathbf{e}_z \\
\mathbf{F}_{\mathrm{a}} - m_{\mathrm{k}}\, g\, \mathbf{e}_z \\
-\mathbf{v}_1^\top \mathbf{v}_1 + \frac{1}{N^2}\left( \dot{l}_{\mathrm{t}}^{\,2} + l_{\mathrm{t}} \ddot{l}_{\mathrm{t}} \right) \\
-\left( \mathbf{v}_{\mathrm{kcu}} - \mathbf{v}_1 \right)^\top \left( \mathbf{v}_{\mathrm{kcu}} - \mathbf{v}_1 \right) + \frac{1}{N^2}\left( \dot{l}_{\mathrm{t}}^{\,2} + l_{\mathrm{t}} \ddot{l}_{\mathrm{t}} \right) \\
-\left( \mathbf{v}_{\mathrm{k}} - \mathbf{v}_{\mathrm{kcu}} \right)^\top \left( \mathbf{v}_{\mathrm{k}} - \mathbf{v}_{\mathrm{kcu}} \right)
\end{bmatrix}
\tag{14}
$$

in which

$$
\mathbf{G}_X = \begin{bmatrix}
\mathbf{r}_1 & \mathbf{0}_{1\times3} & \mathbf{0}_{1\times3} \\
(\mathbf{r}_1 - \mathbf{r}_{\mathrm{kcu}})^\top & (\mathbf{r}_{\mathrm{kcu}} - \mathbf{r}_1)^\top & \mathbf{0}_{1\times3} \\
\mathbf{0}_{1\times3} & (\mathbf{r}_{\mathrm{kcu}} - \mathbf{r}_{\mathrm{k}})^\top & (\mathbf{r}_{\mathrm{k}} - \mathbf{r}_{\mathrm{kcu}})^\top
\end{bmatrix},
\tag{15}
$$

$\mathbf{I}_3$ is the identity matrix, $\mathbf{F}_{\mathrm{a}}$ is the aerodynamic force acting on the wing, $g$ is the gravitational constant, and $\mathbf{e}_z = [0\ 0\ 1]^\top$. The equations of motion for the point masses are described in the upper three rows. The constraint equations described in the lower three rows represent the rigid links between the point masses.

The constraint equations in the lower three rows of Eq. (14) are inferred from the constraints on the distances between linked point masses. The distance between the control unit and the top wing surface point masses is constrained by the constant bridle length $l_{\mathrm{b}}$

$$
c_{\mathrm{b}} = \frac{1}{2}\left( (\mathbf{r}_{\mathrm{k}} - \mathbf{r}_{\mathrm{kcu}})^\top (\mathbf{r}_{\mathrm{k}} - \mathbf{r}_{\mathrm{kcu}}) - l_{\mathrm{b}}^2 \right) = 0.
\tag{16}
$$

The relative distances between the remaining linked point masses are constraint by the instantaneous tether length $l_{\mathrm{t}}$

$$
c_1 = \frac{1}{2}\left( \mathbf{r}_1^\top \mathbf{r}_1 - \left( \frac{l_{\mathrm{t}}}{N} \right)^2 \right) = 0
\tag{17}
$$

and

$$
c_2 = \frac{1}{2}\left( (\mathbf{r}_{\mathrm{kcu}} - \mathbf{r}_1)^\top (\mathbf{r}_{\mathrm{kcu}} - \mathbf{r}_1) - \left( \frac{l_{\mathrm{t}}}{N} \right)^2 \right) = 0.
\tag{18}
$$

These constraints are differentiated twice to yield an index-1 DAE, enabling more efficient integration. As a consequence of the index reduction, the tether length acceleration and the accelerations of the point masses appear in the constraint equations. Moreover, the initial states must be chosen such that they ensure consistent kinematics of the tether and point masses in the simulation. As such, the initial states must satisfy two consistency conditions for each constraint. The original expressions for the constraints provide the first condition. The time derivatives of these expressions provide the second condition

$$
\dot{c}_{\mathrm{b}} = (\mathbf{r}_{\mathrm{k}} - \mathbf{r}_{\mathrm{kcu}})^\top (\mathbf{v}_{\mathrm{k}} - \mathbf{v}_{\mathrm{kcu}}) = 0,
\tag{19}
$$

$$\dot{c}_1 = \mathbf{r}_1^\top \mathbf{v}_1 - \frac{l_t \dot{l}_t}{N^2} = 0 \tag{20}$$

and

$$\dot{c}_2 = (\mathbf{r}_{kcu} - \mathbf{r}_1)^\top (\mathbf{v}_{kcu} - \mathbf{v}_1) - \frac{l_t \dot{l}_t}{N^2} = 0. \tag{21}$$

To prevent inaccuracies of an aerodynamic model of the wing from interfering with the simulation, we do not resolve the dynamics of the point mass of the wing. Instead, the acceleration of the wing is prescribed and used as input. The wing acceleration is inferred from a cross-wind flight path from the flight data, as described in Appendix A. Consequently, the equation of motion of the wing in the third row of Eq. (14) becomes redundant and is dropped for this analysis

$$\begin{bmatrix} \begin{bmatrix} m_1 \mathbf{I}_3 & \mathbf{0}_{3\times3} \\ \mathbf{0}_{3\times3} & m'_{kcu} \mathbf{I}_3 \end{bmatrix} & {\mathbf{G}'}_X^\top \\ \mathbf{G}'_X & \mathbf{0}_{3\times3} \end{bmatrix} \begin{bmatrix} \mathbf{a}_1 \\ \mathbf{a}_{kcu} \\ \lambda_1 \\ \lambda_2 \\ \lambda_b \end{bmatrix} = \begin{bmatrix} \mathbf{D}_{t,1} - m_1 g \mathbf{e}_z \\ \mathbf{D}'_{kcu} - m'_{kcu} g \mathbf{e}_z \\ -\mathbf{v}_1^\top \mathbf{v}_1 + \frac{1}{N^2}\left(\dot{l}_t^{\,2} + l_t \ddot{l}_t\right) \\ -(\mathbf{v}_{kcu} - \mathbf{v}_1)^\top (\mathbf{v}_{kcu} - \mathbf{v}_1) + \frac{1}{N^2}\left(\dot{l}_t^{\,2} + l_t \ddot{l}_t\right) \\ -(\mathbf{v}_k - \mathbf{v}_{kcu})^\top (\mathbf{v}_k - \mathbf{v}_{kcu}) - (\mathbf{r}_k - \mathbf{r}_{kcu})^\top \mathbf{a}_k \end{bmatrix} \tag{22}$$

in which $\mathbf{G}'_X$ is Eq. (15) with the third column removed. Moreover, the term with the wing acceleration in the algebraic equation of the kite element is moved to the right-hand side.

Incorporating the accelerations of the point masses, except for the wing point mass, as algebraic states allows the DAE of the full model to be expressed in a semi-explicit form. The time derivatives of the differential states are

$$\dot{\mathbf{x}} = [\mathbf{v}_1, \, \mathbf{v}_{kcu}, \, \mathbf{v}_k, \, \mathbf{a}_1, \, \mathbf{a}_{kcu}, \, \mathbf{a}_k, \, \dot{l}_t, \, \ddot{l}_t] \tag{23}$$

and Eq. (22) provides the algebraic equations. The DAE is solved with the IDAS integrator in CasADi (Andersson et al., 2019). IDAS employs the backward differentiation formula (variable-order, variable-coefficient) for implicit integration to solve the system. The motion is resolved at a fixed time step of 0.1 s. The solver produces a consistent simulation with insignificant drift in the consistency conditions, i.e., the distance between the wing and the KCU drifts with 0.0001 m in 24.2 s.

In contrast to the steady-rotation state calculation in Sec. 3.2, drag is calculated directly with the local apparent wind velocity $\mathbf{v}_{a,j}$ instead of its normal component $\mathbf{v}_{a\perp,j}$ (Eqs. (5), (6), and (7)) to limit the non-linearity of the model. To sum up, we incorporate the following model modifications with respect to the work of Zanon et al. (2013):

– The tether length time derivatives are added to the dynamic equations to enable modelling pumping AWE systems;

– Drag is computed directly at the point masses instead of being computed at the centres of the tether elements and then lumped to the adjacent point masses;

– The acceleration of the wing point mass is not solved. Instead, the wing acceleration inferred from measurements is directly imposed;

– We add an extra element (rigid link) to represent the kite as described in Sec. 3.1.

## 4   Results

Firstly, the steady-rotation-state approximation is used to study the motion of the tether and kite along the figure-of-eight manoeuvre. A discretisation by 30 tether elements is compared with a minimal discretisation using only a single tether element. Secondly, the motion is simulated with the dynamic model using 30 tether elements. Subsequently, the resulting roll and pitch along the figure of eight from the different models are compared with measurements. Finally, the motion of the tether and kite along a full pumping cycle is studied.

### 4.1   Tether-kite lines computed with steady-rotation states

The steady-rotation-state approximation uses the measured tether force, wing position, and optimised angular velocity to determine the instantaneous positions of the point masses. The line formed by the elements between these point masses is referred to as the tether-kite line. Figure 9 shows the resulting tether-kite lines with 30 tether elements at the reference instances.

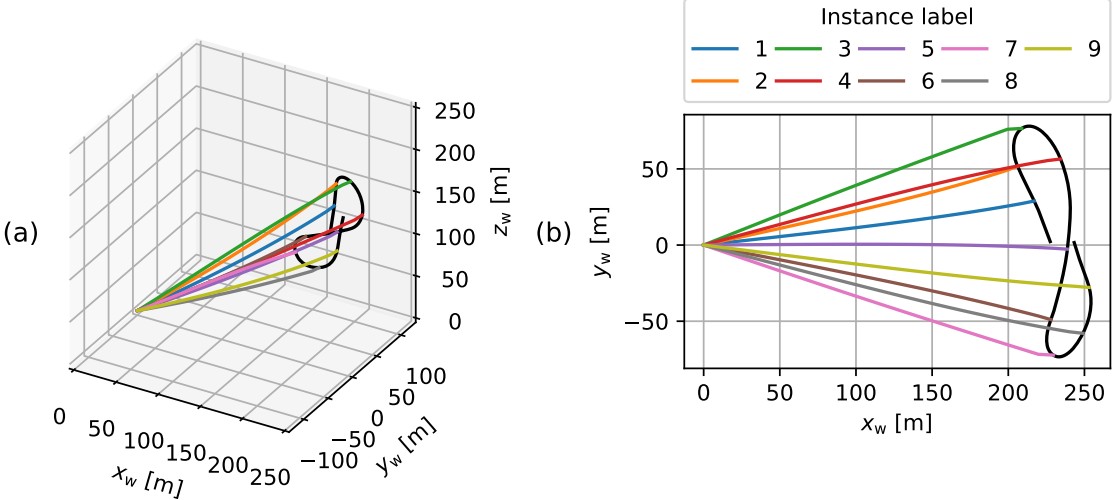

**Figure 9.** Tether-kite lines for the nine reference instances resulting from the steady-rotation-state approximation with the tether discretised by 30 elements in 3D (**a**) and top-view (**b**).

Variations in the deformation of the tether-kite line are hard to identify with the naked eye in the previous plots. Therefore, the cross-axial displacement is plotted against the radial position for the first five reference instances with the solid lines in Fig. 10. The displacement is expressed with respect to the tangential apparent wind velocity of the kite. The largest displacements are found in the down-apparent-wind direction, which can be attributed to the tether drag. The direction in which gravity contributes to the displacement varies depending on the position along the figure-of-eight manoeuvre. Table 3 specifies in which direction gravity acts for the first five reference instances. For all instances except for the third, gravity acts in the down-apparent-

wind direction. The cross-apparent-wind displacement contribution of gravity changes sign after the third instance. Finally, the resistance to turn, or the inertia, mostly contributes to the displacement in the positive cross-apparent-wind direction, as can be inferred from the high positive values in the last column of Table 3.

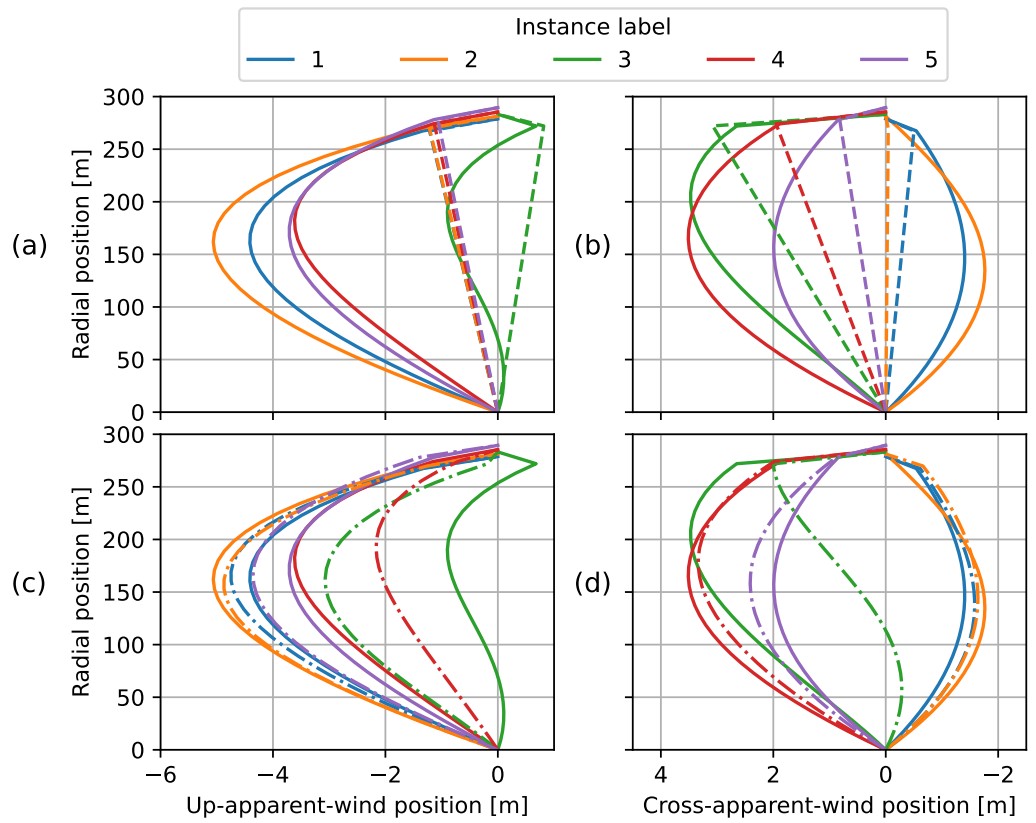

**Figure 10.** Tether-kite lines with cross-axial displacement decomposed with respect to the tangential apparent wind velocity of the wing (see Fig. 5). Steady-rotation states with 30 tether elements (solid lines in **a**, **b**, **c**, and **d**), with a single tether element (dashed lines in **a** and **b**), and the dynamic solution with 30 tether elements (dash-dotted lines in **c** and **d**) for the first five reference instances. Note that the x- and y-axes have different scales and that the x-axes are flipped in the second column.

The discontinuities in the tether-kite lines at the KCU indicate that it has a substantial effect on the attitude of the kite element. The high mass and drag lumped to the KCU point relative to the mass and drag lumped to the tether points cause these discontinuities.

     To investigate the imposed kite attitude more precisely, it is quantified using the pitch and roll of the kite element with respect to the tangential plane (perpendicular to the position vector of the kite). The corresponding transformation is described

in more detail in Appendix B. Figure 11a shows that the pitch is roughly constant during the straight flight path sections and drops below zero during the turns (blue line). The negative pitch is confirmed by the tether-kite line plot of the 3$^{\text{rd}}$ instance in

**Table 3.** The negated vertical unit vector $-\mathbf{e}_z$ and the negated centripetal unit vector $-\mathbf{e}_{\text{centripetal}}$ decomposed in the up-apparent-wind and cross-apparent-wind direction experienced by the wing. The centripetal unit vector is determined by the approximated centripetal acceleration at the kite $\mathbf{e}_{\text{centripetal}} = \frac{\boldsymbol{\omega}_{\text{turn}} \times (\boldsymbol{\omega}_{\text{turn}} \times \boldsymbol{r}_{\text{k}})}{\|\boldsymbol{\omega}_{\text{turn}} \times (\boldsymbol{\omega}_{\text{turn}} \times \boldsymbol{r}_{\text{k}})\|}$. The listed fractions help to explain the contributions of gravity and turn inertia to the cross-axial displacement of the tether-kite lines in Fig. 10.

| Instance label | $-\mathbf{e}_z$ | | $-\mathbf{e}_{\text{centripetal}}$ | |
| --- | --- | --- | --- | --- |
| | Up | Cross | Up | Cross |
| 1 | -0.56 | -0.55 | 0.09 | 0.41 |
| 2 | -0.44 | -0.62 | 0.28 | 0.95 |
| 3 | 0.72 | -0.32 | 0.25 | 0.97 |
| 4 | -0.03 | 0.84 | -0.23 | 0.96 |
| 5 | -0.46 | 0.68 | -0.27 | 0.84 |

**Figure 11.** The pitch and roll of the kite derived from the attitude of the kite element (with respect to the tangential plane) along the figure of eight. The results of the steady-rotation-state and dynamic analyses are depicted alongside the pitch and roll inferred from attitude measurements of the two sensors mounted to the wing, which include local effects of wing deformation. The shaded intervals indicate the turns.

Fig. 10a, where the upper kite element is tilted backwards. Note that this observation is specific to the instantaneous direction of the apparent wind velocity. It may seem that the KCU is leading the wing during the downwards-flying turn; however, it is actually positioned higher above the ground.

Figure 11b shows a distinct pattern for the roll of the kite along the figure of eight (blue line). The roll is nearly constant and slightly negative at the first straight section flying to the left, whereas it is slightly positive at the subsequent straight section flying to the right. In between, during the left turn, the roll peaks in the middle of the turn at 36.2 s. The right turn shows an opposite pattern. The rolling motion of the kite during the turns can be predominantly attributed to the resistance to turn, or inertia, of the KCU. The inertia of the tether has a much smaller effect on the roll. This stresses the need for including a separate point mass for the KCU when assessing the kite attitude.

The analysis is repeated using a single tether element. Figure 10a and b show the resulting tether-kite lines with the dashed lines. As expected, this minimal model is not able to give a good estimation of the maximum displacements. Nevertheless, the computed pitch and roll of the kite element are similar for both discretisations, as can be observed in Figure 11.

## 4.2   Cross-check with dynamic results

The dynamic simulation requires the wing acceleration, imposing the flight path, and the tether reel-out acceleration as input. The flight trajectory is reconstructed as described in Appendix A to enable a running simulation and ensure that the inputs are consistent with the studied figure-of-eight manoeuvre. The intensive reconstruction yields a slightly adapted tether reel-out speed with respect to the measured speed and imposes a nearly constant difference between the tether length and radial kite position in the simulation. In this paper, we refer to this difference as the tether slack. The initial tether length of the simulation is chosen such that the tether slack is 0.28 m, which is the mean value observed in the steady-rotation-state results.

Figure 12 shows the tether force evolution that results from the dynamic simulation. The agreement with measurements during the straight sections confirms that the choice for the constant tether slack is reasonable. During the turns, the calculated tether force does not agree well with the measurements. The simulated force shows distinct peaks, whereas the measured force shows a more gradual increase. These differences are not specific to the dynamic model and are expected to be artefacts of the wing and tether acceleration control input. The wing control input being a source of error is affirmed by coinciding, unexpected tether length results computed with the steady-rotation states. Consequently, also the tether acceleration control input will be a source of error since it is derived from these results. Errors introduced by model deficiencies, such as neglecting kite deformation and elasticity and damping of the tether, are expected to be overshadowed by relatively high errors due to the input.

The resulting tether-kite lines are plotted in Fig. 10c and d. Most shapes of the reference instances show a reasonable agreement with the steady-rotation-state results. An apparent outlier is the 3rd reference instance, which occurs at the outside of the turn. This discrepancy can also be observed in Fig. 11a, in which the pitch resulting from the dynamic simulation follows the steady-rotation-state results during the straight flight sections but not during the turns.

The transient effects on the tether-kite line that arise from the highly dynamic flight behaviour during turns are accounted for by the dynamic model but not by the steady-rotation states. This explains why in Fig. 10d, the lower end of the tether

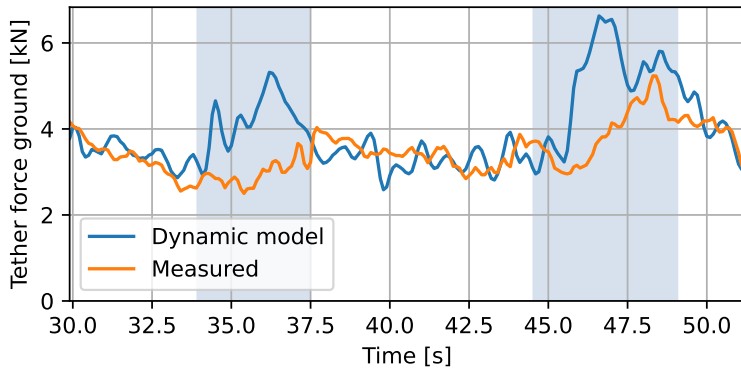

**Figure 12.** Tether force evolution along the figure of eight resulting from the dynamic simulation and from the flight data. The shaded intervals indicate the turns.

of the 3$^{rd}$ reference instance has a negative cross-apparent-wind displacement like its predecessor, while the corresponding steady-rotation-state result is positive over the full length.

Despite including transient effects, the dynamic model does not necessarily enhance accuracy as it requires significant assumptions, e.g., for acquiring the tether reel-out acceleration input. Moreover, the dynamic simulation is expected to be more sensitive to neglecting tether elasticity and damping. One aspect that demonstrates this sensitivity is the relatively large oscillations observed in the pitch and roll of the kite computed with the dynamic model.

### 4.3 Kite attitude validation

The wing attitude measurements enable estimating the pitching and rolling motion of the kite assembly and, thereby, can be used to validate the computed results. Validating the rotational motion of the kite is particularly important for performance model development, as accurate descriptions of this motion are essential for incorporating the aerodynamics and the turning mechanism. The tether motion cannot be validated as no measurements are taken directly from the tether.

Figure 11 compares the computed pitch and roll angles inferred from the kite element orientation with measurements from the two different sensors mounted to the inboard struts of the wing. The same pitch and roll definitions are used to express the wing attitude measurements, provided in Appendix B. The kite attitude is inferred from these measurements by assuming that the kite is rigid and that the orientation of the wing relative to the bridle is defined by the depower angle $\alpha_{\mathrm{d}}$ shown in Fig. 3. Moreover, the measurements are corrected for misalignments with the wing reference frame. 7° is added to the measured pitch of both sensors to correct for the sensor misalignment. Similarly, 8.5° is subtracted from the roll of both sensors to correct for sensor misalignment.

Both sensors measure a similar roll along the whole figure of eight, as shown in Fig. 11. However, the pitch measured with the two sensors differs substantially during the turns. This difference can be attributed to the steering input. A steering input causes the steering tape to pull in on one side and give slack on the other. As a result, the wing tip section at the inside of

the turn will locally pitch nose-up while the wing tip section at the outside of the turn will pitch nose-down. This behaviour can be observed in Fig. 14. The left wing tip of the wing turning left (Fig. 14b) slightly pitches nose-up with respect to the wing without steering (Fig. 14a). Similarly, the right wing tip of the wing turning right (Fig. 14c) slightly pitches nose-up with respect to the wing without steering. Investigating the relationship between the pitch differential of the two inboard ribs and the steering input shows a high correlation. Figure 13c illustrates this relationship for the 65th pumping cycle, which exhibits a Pearson correlation coefficient of -0.96. This relationship indicates that steering-induced wing twisting is being measured: a steering input makes the wing twist around the leading edge along the whole span, with a zero twist at the centre. The high correlation strength suggests that the twist between the struts on which the sensors are mounted is measured with high precision. The pitch at the centre of the wing is assumed to be the average of the two measurements.

Figure 11 shows that the differences in pitch and roll resulting from the models and the measurements are small during the straight sections where there is no steering-induced wing twisting occurring. The computed pitch and roll angles match the measurements within three degrees. Although the dynamic result lies closer to the average measured pitch during the turns, it does not exhibit a similar peak. This discrepancy is expected to be caused by multiple sources of error. First, the actual wing motion that is causing the peak in pitch during the turns might not be accurately reconstructed in the flight trajectory reconstruction. This is affirmed by the large imposed modifications due to the relatively high uncertainty of the position measurement during the turns. Second, the pitch of the kite assembly calculated with the rigidly-linked two-point kite model is inadequate to describe the desired pitching behaviour of the kite during this highly dynamic manoeuvre. Consequently, a higher-fidelity model might be needed to obtain a suitable, higher-resolution pitching motion description. And third, the available measurements could be insufficient to estimate the pitch of the kite assembly due to the measurement of local wing deformation. This would lead to discrepancies even if the kite model would be adequate.

The possible inaccuracy of the kite model structure in highly dynamic states pertains to the assumption that the kite assembly (consisting of the wing, bridle, and KCU) is rigid. In reality, the kite deviates substantially from the CAD geometry in Fig. 3, even in steady states. For example, the canopy will billow towards the trailing edge, as shown in Fig. 1. Figure 14 shows video stills obtained with a KCU-mounted camera which records how the kite assembly deforms during the turns. If the kite were rigid, the wing would have a fixed position in the field of view of the camera. However, during the turns, the wing shifts sideways towards the turning centre. This shift is caused by the changing bridle and wing geometry due to steering actuation, asymmetric aerodynamic loading, and inertial effects. The shift increases substantially along the turning manoeuvre and reaches a maximum at the outside of the turns, where the kite speed is high, and the turning radius is the smallest. Further research is needed to investigate if patterns observed in the measured pitch and roll along the figure-of-eight manoeuvre can be attributed to these kite deformations.

In general, the steady-rotation states perform reasonably well in estimating the kite attitude, both with a single tether element and 30 tether elements. This suggests that the coarse discretisation is equally effective in capturing the inertial effect of the KCU during turns. Despite including transient effects, the dynamic model does not necessarily show a better agreement with measurements than the steady-rotation-state model. This suggests that improving the method for solving the motion may not

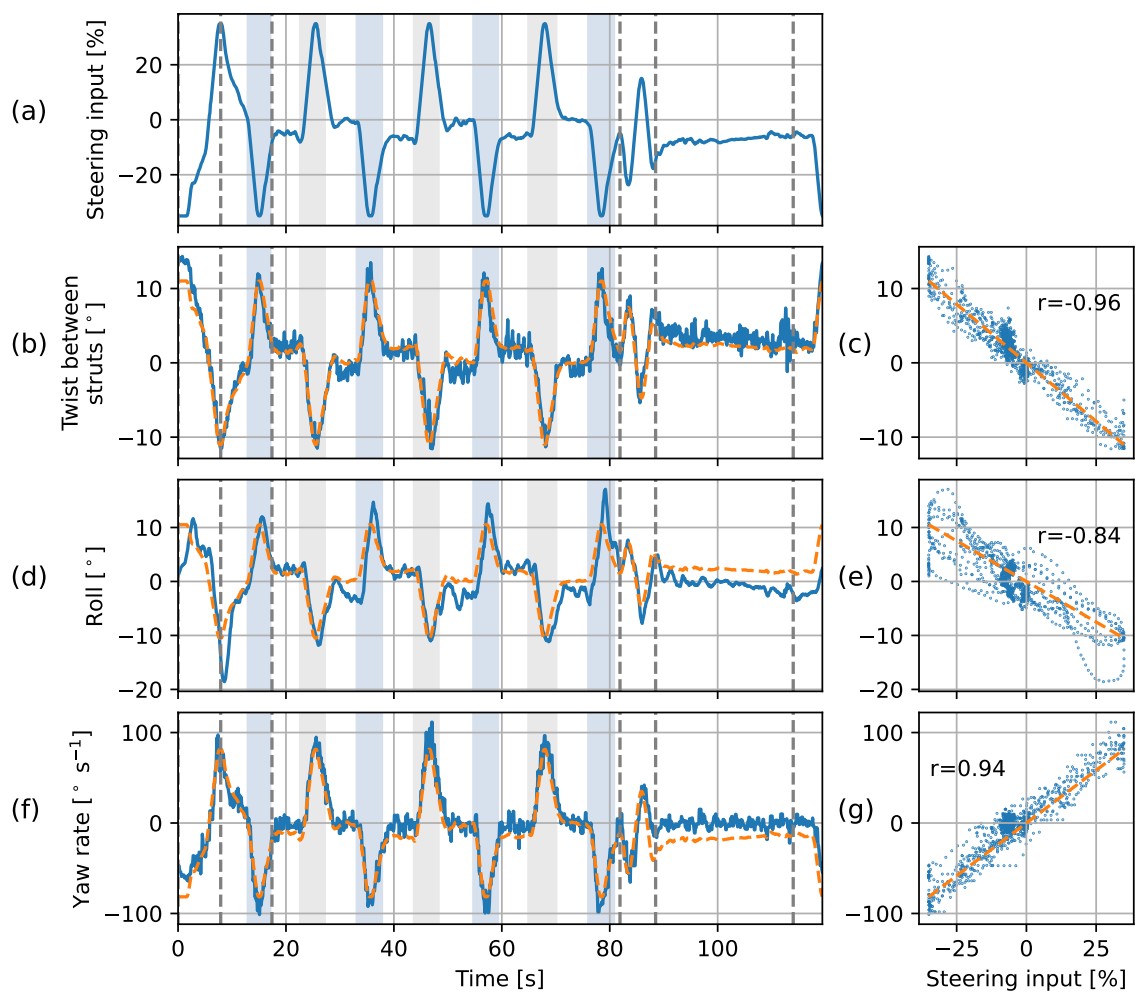

**Figure 13.** Relations between (**a**) the steering input and (**b, c**) the difference in pitch of the two sensors, (**d, e**) roll of the kite, and (**f, g**) yaw rate of the kite in the 65<sup>th</sup> pumping cycle. The orange dashed lines in the left column depict the steering input scaled with the slope found in the linear fit shown with the orange dashed lines in the right column.

be effective unless the configuration of the kite model itself is refined to compute pitching motion more accurately. However, a definite conclusion cannot be drawn because the uncertainty of the measurements might distort the view of the model validity.

### 4.4 Pitching motion along a full pumping cycle

To study the pitching motion of the kite outside the reel-out phase, we zoom out and evaluate multiple pumping cycles, including the 65<sup>th</sup> cycle, which contains the previously investigated figure-of-eight manoeuvre. During the reel-in phase, the kite only requires small steering adjustments. Consequently, the kite does not show significant rolling. In contrast, the pitching

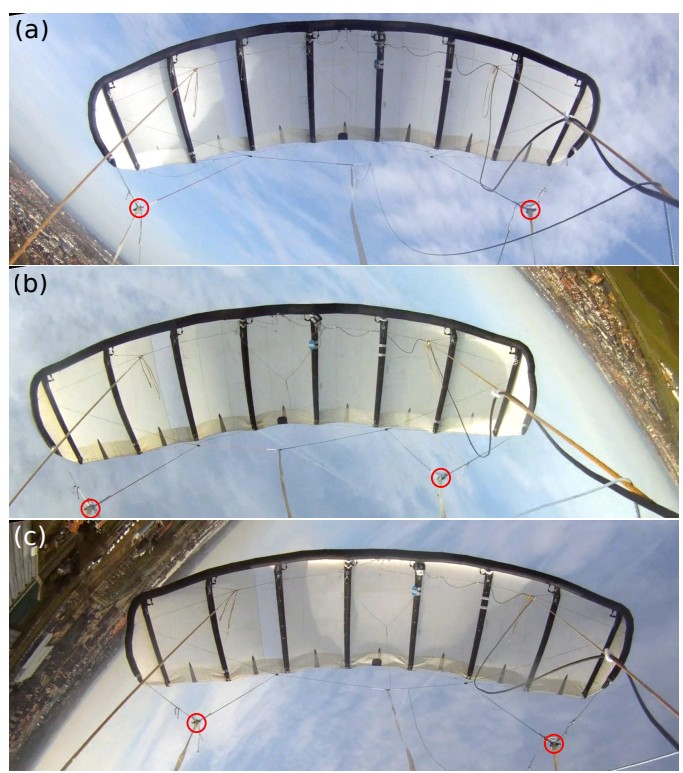

**Figure 14.** Video stills obtained from a recording from a test flight on 30 March 2017 with a kite largely identical to the V3.25B kite and a GoPro® camera mounted to the KCU. (**a**) Kite flying on a straight path just after launch. (**b**) Kite performing a left turn and located between reference positions 3 and 4. (**c**) Kite performing a right turn and located between reference positions 7 and 8. Reference positions refer to Fig. 5. All stills were cropped in exactly the same way from the original recording and rotated by -9° to correct for the misaligned mounting on the KCU. The overlaid red circles indicate the positions of the steering tape endpoints, visualising the steering-induced asymmetry of the bridle line system.

of the kite increases due to the increased tether sag. The increased sag results from a decrease in tether tension, which makes the weight and drag of the tether more dominant.

Figure 15 shows the kite pitch inferred from the wing measurements and the kite pitch resulting from the steady-rotation-
state analysis with 30 tether elements. The results of ten consecutive pumping cycles are depicted, starting with the 65[th] pumping cycle. Each cycle starts with the transition into the reel-out phase, followed by approximately four figures of eight. Subsequently, the kite is pointed towards the zenith, depowered, and reeled back in (after the last shaded interval). The cycle ends after powering up again in preparation for a new cycle.

Each cycle shows an increase in pitch after the last turn in the reel-out phase as the kite transitions into the reel-in phase.
The model overestimates the pitch at the start of the reel-in and underestimates it towards the end but gives a good overall agreement. There are many factors that may cause this discrepancy. One plausible explanation is that the reduced load during

the reel-in phase leads to the deformation of the kite struts on which the sensors are mounted. The deformation is measured but not accounted for in the model and, thus, not incorporated in the computed results. Note that during the reel-in, the steering input is non-zero, as shown in Fig. 13a. This causes a pitch offset between the two sensors.

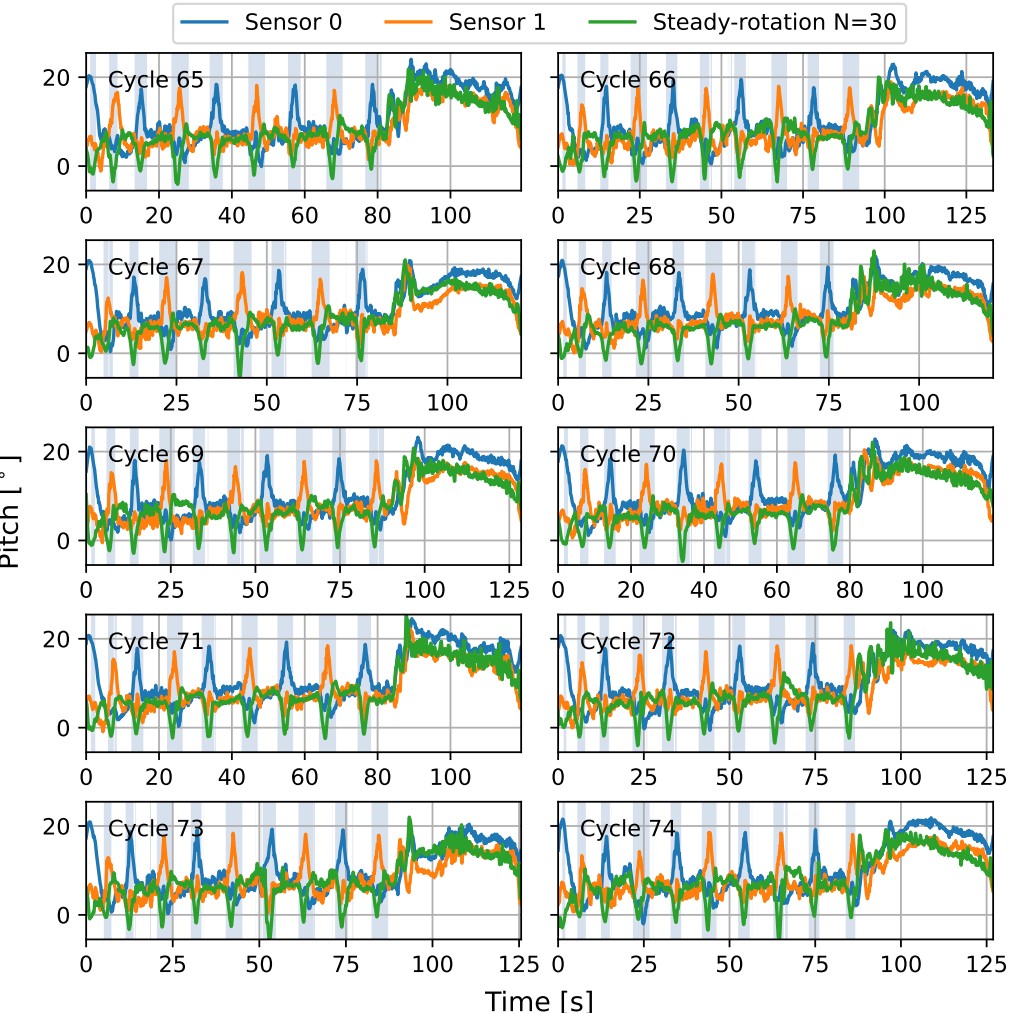

**Figure 15.** The pitch of the kite element with respect to the tangential plane along ten pumping cycles resulting from the steady-rotation-state analysis using 30 tether elements (T-I N=30), together with the kite pitch inferred from the wing attitude measured with two sensors. The shaded intervals indicate the turns during the reel-out phase. After the turns, the system transitions into the reel-in phase.

## 5   Discussion

In this section, we discuss the turning mechanism and the implications of the observed swinging motion for the performance modelling of a kite system. Different mechanisms initiate and drive a turn of a flexible kite system with a suspended control unit.

The initiation mechanism relies on twisting the wing tips, as discussed in Sec. 4.3. A steering input induces twisting of the wing, leading to changes in the pitch of different wing sections along the span. This increases the angle of attack at the wing tip at the inside of the turn and decreases it at the outside wing tip. This creates an aerodynamic side force component perpendicular to the kite symmetry plane and pointing towards the turn centre. The introduction of a side component effectively rolls the resultant aerodynamic force acting on the whole kite without rolling the kite itself. In contrast to flexible kites with a suspended control unit, multi-line flexible kites that are actuated from the ground employ this mechanism to drive the whole turn; the side force is dominant in providing the centripetal force.

The driving mechanism for turning flexible kites with a suspended control unit is the rolling of the kite. As soon as the turn is initiated, the kite will roll into the turn to exert a centripetal force on the relatively heavy KCU, pulling it along. Together with the kite, the lift force generated by the top wing surface rolls into the turn and contributes to the centripetal force. The higher the mass of the KCU, the more roll is required to execute the same turn. Consequently, a smaller fraction of the lift is available to carry the weight of the airborne components and pull the tether. While the aerodynamic side force is still necessary to maintain turning, it is the roll of the kite that accommodates the largest contribution to the centripetal force and is thus considered to drive the turn.

To incorporate this turning mechanism, a single-point kite model would need the roll of the kite as an input, relying on the user to provide realistic roll angles. Another option is modelling the roll, e.g., using an empirical relationship between the roll and the steering input, as shown in Fig. 13e. However, with little extra computational cost, the roll can be resolved by modelling the kite with at least two point masses: one for the wing and one for the KCU. Thereby, it no longer needs to rely on system-specific empirical relationships to include the steering mechanism. Instead, the aerodynamic side force needed to initiate and maintain the turn can be calculated based on the deformation of the kite tips and associated aerodynamics.

Although the kite pitch does not change substantially during the reel-out phase, the tether-kite motion changes the pitch substantially outside this phase. The sag-induced pitch concerns performance modelling as it affects the angle of attack experienced by the wing, which in turn affects the generated aerodynamic forces. Resolving the pitch also requires modelling the kite with at least two point masses and enables incorporating an aerodynamic model for the wing with a dependency on the angle of attack.

Given the coarse discretisation of a two-point representation of the kite, it is reasonable to adopt a rigid-kite assumption when employing this kite model. The adequacy of using a two-point kite model for performance modelling pertains to the validity of the rigid-kite assumption. Despite that the kite assembly is subject to a changing geometry, the errors arising from the rigid-kite assumption are expected to be limited. Therefore, this assumption can be justified by the necessity to limit computational complexity as required for some types of analysis for which the performance model can be useful.

Employing the developed models for performance modelling requires that the model is complemented with an aerodynamic model of the wing. This enables the dynamic model to resolve the flight path and a quasi-steady model to compute the kite speed along a partially prescribed flight path. Note that incorporating the wing aerodynamics makes the models much more sensitive to the wind input.

## 6 Conclusions

The inertia of the suspended control unit has a large effect on the roll of a flexible kite during turns in the reel-out phase. During the reel-in phase, the pitch of the kite changes due to the weight and drag of the control unit and increased tether sag. These effects are not resolved when the kite is modelled with a single point mass. With two point masses, one at the wing and one at the control unit, the steady-rotation-state model performs reasonably well in capturing the pitch and roll with little extra computational effort. A two-point model of the kite can thus be a powerful tool for the performance modelling of flexible kite systems.

The swinging motion of a kite with a suspended control unit is assessed with two approaches: approximated as a transition through steady-rotation states and solved dynamically. In contrast to the dynamic model, the steady-rotation-state model neglects transient effects. Both approaches employ a two-point kite model extending a discretised tether model using an additional rigid element for the kite. By prescribing the cross-wind flight path of the wing, no aerodynamic model of the kite is required.

An alternative expression for the angular velocity underlying the steady-rotation assumption is derived that accounts for the turning of the kite. This angular velocity expression accommodates lateral accelerations on the point masses and, thereby, allows studying the lateral swinging motion of the kite. The angular velocity for turns is approximated with flight data and shows good agreement with the kite kinematics. Unlike the original angular velocity expression, the proposed expression yields a good approximation of not only the wing velocity but also of the wing acceleration.

The tether-kite lines resulting from the steady-rotation states show discontinuities at the junction between the tether and the kite. These indicate that the control unit has a substantial effect on the attitude of the kite and stress the need for including a separate point mass for the control unit in performance models for flexible kite systems. The steady-rotation states perform reasonably well in estimating the roll of the kite, both with a single and 30 tether elements. The computed pitch and roll angles match the measured angles within three degrees during the straight sections of the figure-of-eight manoeuvre. During the turns, the peaks in the roll are overestimated, and the instantaneous differences in roll may exceed five degrees, whereas the pitch exhibits more systematic differences. These systematic differences could partially be explained by the fact that the model did not account for transient effects. However, drawing a definite conclusion is challenging, as the measurements include steering-induced pitch, making the wing measurements a poor reference.

Although the dynamic model considers transient effects, it does not prove to be more accurate in capturing the roll and pitch behaviour during turns than the steady-rotation states. This is expected to be primarily caused by inaccuracies in the wing acceleration and tether reel-out acceleration inputs. Due to anomalies in the flight trajectory measurements, a reconstruction was

necessary to generate consistent inputs, enabling a running simulation. The reconstruction assumes that the tether slack length, defined as the difference between the tether length and radial position of the kite, remains constant. The large modifications imposed by the reconstruction add further uncertainty to the results. Moreover, since the developed model aims for simplicity to increase computational efficiency, it does not incorporate all relevant mechanical effects, such as tether elasticity and damping. In addition to solving the motion dynamically, it could be necessary to refine the configuration of the kite model in order to increase the accuracy of solving the pitching motion and explain the observed differences between the measured and computed pitch.

Two separate mechanisms have been identified that initiate and drive a turn of a flexible kite system with a suspended control unit. A steering input causes an aerodynamic side force that initiates the turn. As soon as the turn is initiated, the kite starts to roll as it needs to pull the relatively heavy control unit into the turn. The rolled lift force provided by the top wing surface of the kite provides the largest contribution to the centripetal force and is said to drive the turn. Since a two-point kite model resolves the roll, the lift force may tilt along with the kite to drive turns. Hence, it avoids making large assumptions to model the centripetal force, as seen in a single-point kite model. Furthermore, by resolving the pitch, the kite model allows computing the angle of attack of the wing, which is crucial for obtaining an accurate aerodynamic model. This becomes particularly important when solving the wing motion instead of prescribing a flight path, as done in the current study. Further study is needed to assess how refined the pitching motion needs to be solved to accurately calculate the angle of attack of the wing.

The results of this study could be significantly improved with better quality flight data, more raw data, and information about how measurements are conditioned and calibrated. Currently, the sensor units are mounted to the flexible wing. As a result, wing deformation and actuation of the depower angle of the wing are also measured. This could be prevented by mounting the sensor units to the kite control unit. To find a better match between the measured and simulated tether forces, it would be interesting to incorporate variable tether slack and account for stretching in the dynamic simulation. A stepping stone could be to wrap the simulation in an optimisation problem to find the tether acceleration input that produces the measured tether force and cross-check the results with the tether lengths resulting from the steady-rotation states. More accurate tether length information in the experimental data would greatly help such analysis. Moreover, the flight trajectory reconstruction could be enhanced with this information, as well as with more advanced state estimation techniques. Finally, both the steady rotation state and dynamic model could still benefit from refining the wind modelling and fine-tuning the model parameters.

*Code and data availability.* The complete test flight data, including 87 pumping cycles spanning a total flight time of 265 minutes, are available in open access from (Schelbergen et al., 2024). The specific pumping cycle underlying this study and the Python code for the data analysis are available in open source from (Schelbergen, 2024).

## Appendix A: Flight trajectory reconstruction

The kinematics of the wing recorded in the flight data show inconsistencies in the measured tether reel-out speed and are reconstructed in a preprocessing step to remove anomalies. The dynamic simulation relies on the recorded wing kinematics and tether reel-out speed for its input. Directly using these recorded quantities as input leads to faulty simulations, and a workaround is needed to obtain coherent input. The reconstruction is carried out for the full 65[th] pumping cycle.

A preliminary evaluation of the wing kinematics in the flight data shows that the vertical speed does not fully agree with the derivative of the vertical position of the wing, even though it does for the horizontal components. The largest mismatch occurs during the turns, where the recorded vertical speed is more negative than the derivative of the vertical position. The recorded vertical position is GPS data enhanced with barometer measurements. However, we expect that the vertical speed was not updated accordingly.

The inconsistent vertical speed leads to a discrepancy between the derivative of the measured radial position $\hat{r}_k$ and the measured radial component of the wing velocity $\hat{v}_{k,r}$, while in theory, they should be the same. These quantities are depicted with the blue and red lines, respectively, in Fig. A1c. The radial component of the wing velocity is calculated with:

$$v_{k,r} = \frac{\mathbf{r}_k \cdot \mathbf{v}_k}{\|\mathbf{r}_k\|},\tag{A1}$$

in which $\mathbf{r}_k$ and $\mathbf{v}_k$ are the position and velocity of the wing, respectively. An objective of the intended flight trajectory reconstruction is to ensure that the updated radial component of the wing velocity and the derivative of the radial position agree.

As an additional check, the derivative of the measured radial position of the wing $\hat{r}_k$ is compared to the measured tether reel-out speed $\hat{\dot{l}}_t$ (dotted black line in Fig. A1c). The derivative of the radial position shows large fluctuations around the tether reel-out speed in the reel-out phase. The magnitude of the fluctuations conflicts with our expectation that the changes in tether slack (difference between the tether length and radial position of the kite) and stretch are small in this phase. Towards the end of the left turns (at the end of the blue intervals), the derivative of the radial position even tends to become shortly negative.

Figure A1a shows how the integrated measured reel-out speed (dotted black line) evolves with respect to the measured radial position of the wing $\hat{r}_k$ (blue line). During the left turns, the inferred tether length increases approximately linearly, while the radial position exhibits subtle local maxima. These local maxima coincide with the large discrepancies between the derivative of the radial position and the tether reel-out speed observed in Fig. A1c. Note that the tether length lines depict the relative lengths with respect to the start of the pumping cycle. The lines need to be shifted up with their initial values to obtain their respective absolute values. Unfortunately, we do not know the absolute tether length as it is not measured directly.

The residual between the inferred tether length and measured radial position $\Delta \hat{l}_t$ is shown in Fig. A1b. During the left turns, the residual changes roughly 2 m (depth of the valley) within a couple of seconds. The corresponding relatively large increase in radial position can partly be attributed to decreased tether slack and increased tether stretch. However, the magnitude of the change is deemed to be too large to be attributed only to changes in these quantities. Note that also here, the line may shift vertically depending on the initial values. As such, we can not draw conclusions based on the magnitude of the residual but

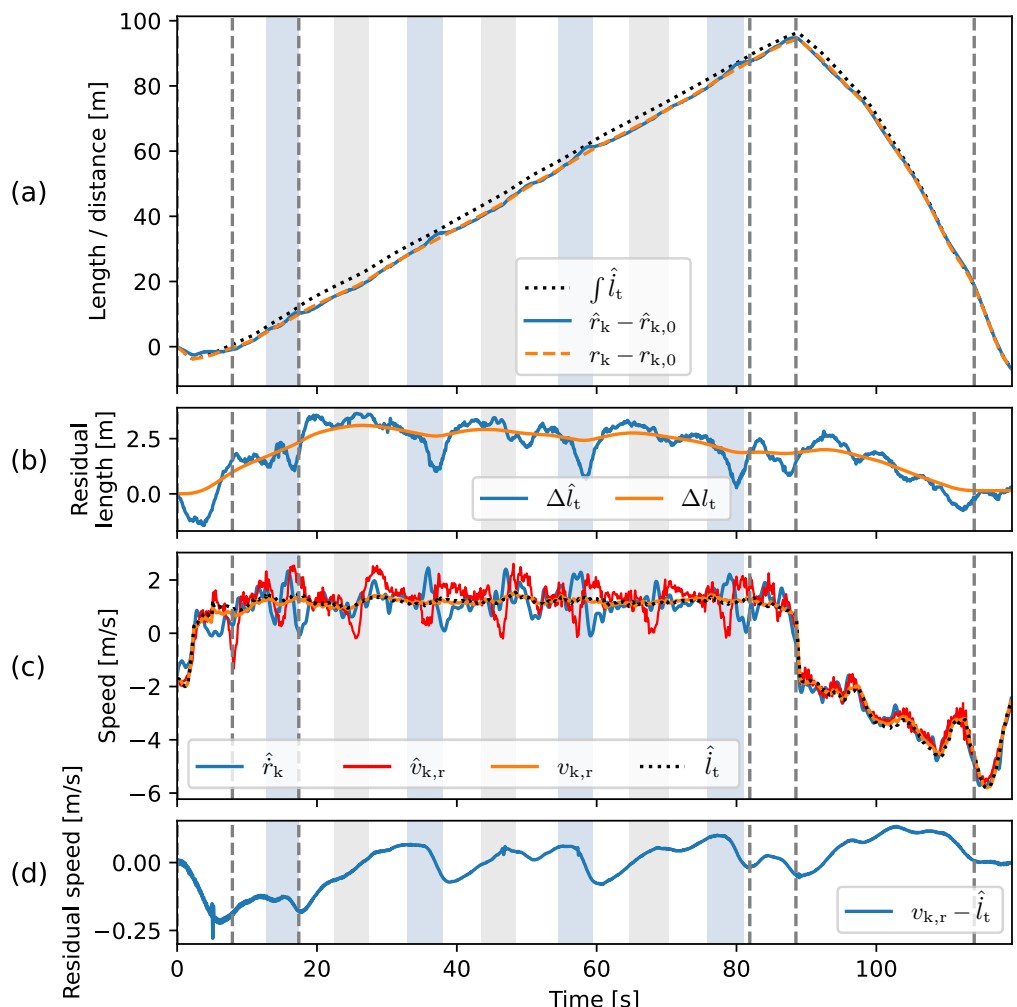

**Figure A1. (a)** Evolution of unstrained tether length $\int \hat{\dot{l}}_t$ and the measured and reconstructed radial distances of the wing, $\hat{r}_k$ and $r_k$, all with their initial values subtracted. **(b)** Difference between the tether length and the measured radial distance of the wing $\Delta \hat{l}_t$ and its equivalent after the reconstruction $\Delta l_t$. **(c)** Time-derivative of measured radial position of the wing $\hat{\dot{r}}_k$, measured and reconstructed radial speeds of the wing, $\hat{v}_{k,r}$ and $v_{k,r}$, and measured tether reel-out speed $\hat{\dot{l}}_t$. **(d)** Residual between the tether reel-out speed and reconstructed radial speed. The intervals shaded grey and blue indicate right and left turns, respectively.

merely on how it changes with time. The given residual length has an unknown offset with respect to the tether slack. Note that
the tether slack cannot be negative.

The maxima in the recorded radial position do not need to be purely physical. Another possible cause is GPS inaccuracy during manoeuvres, which has previously been reported in the literature. Borobia et al. (2018) reported measured radial position

exceeding varying more than 3 m while none was expected. Considering the imprecision of the recorded position, we opt to adapt the wing kinematics by letting the radial wing speed follow the measured reel-out speed as closely as possible.

The flight trajectory reconstruction is obtained using a discrete-time optimisation problem that minimises the error between the modelled radial wing speed and recorded tether reel-out speed while limiting the bias between the modelled and recorded wing position

$$\min_{\mathbf{r}_k(\cdot), \mathbf{v}_k(\cdot), \mathbf{a}_k(\cdot)} \quad \sum_{i=0}^{N} \left[ w \left( v_{k,r} - \hat{\dot{l}}_t \right)^2 + (\mathbf{r}_k - \hat{\mathbf{r}}_k)^\top (\mathbf{r}_k - \hat{\mathbf{r}}_k) \right]_{t=\frac{i}{10}} \tag{A2}$$

$$\text{s.t.} \quad \mathbf{a}_k = \dot{\mathbf{v}}_k = \ddot{\mathbf{r}}_k .$$

Quantities marked with a hat indicate measured quantities, whereas the absence of a hat indicates modelled quantities. A
discrete function is used for the acceleration of the wing, and continuous trajectories are used for the velocity and position of the wing. The decision variables consist of the wing accelerations during the control intervals $\mathbf{a}_k(\cdot)$ and the velocities $\mathbf{v}_k(\cdot)$ and positions $\mathbf{r}_k(\cdot)$ at the control interval boundaries. N is the number of time steps, and the weighing factor $w = 25$ is chosen as it leads to a good balance between the two objectives. Note that having matching reel-out and radial wing speeds does not necessarily mean that also the tether length is the same as the radial position. However, it does mean that the tether slack stays
constant.

In line with the dynamic simulation, the fitting problem uses discrete control input trajectories. It assumes a constant acceleration within each simulation time step of 0.1 s. Between the corresponding control intervals, the values may vary. Due to the step function form of the acceleration, the velocity and position are linear and quadratic functions, respectively, within the control intervals. These low-order forms allow for sufficient detail due to the small time step. The fitting problem is solved in
CasADi using a multiple-shooting approach. This approach is not hindered by integration drift causing an accumulating error with time.

The flight trajectory reconstruction results are shown with the orange lines in Fig. A1. The reconstruction shaves off the local maxima in the recorded radial position, as can be observed in Fig. A1a. Figure A1c shows that the reconstructed radial wing speed follows the measured reel-out speed more closely. The residual speed, which is penalised by the first term of the
objective function, is illustrated in Fig. A1d. The optimiser reduces the position bias, which is penalised by the second term of the objective function, by allowing small changes to the radial wing speed with respect to the measured reel-out speed. As a consequence, the reconstruction does not lower the residual length substantially but keeps it close to the original residual length, as can be seen in Fig. A1b.

We use the reconstructed radial wing acceleration $a_{k,r}$ as tether reel-out acceleration input $\ddot{l}_t$ for the simulation. Thus, we
not only reconstruct the flight trajectory but also modify the tether reel-out speed with respect to the measurements. As a result, the tether slack remains constant in the simulation and is set by the choice for the initial tether length. In reality, changes in slack length will occur, especially during the transition phases. Therefore, this approach might be sub-optimal for simulating the entire pumping cycle. Nonetheless, it is suitable for simulating intervals where only small tether slack and stretch changes are expected, such as the reel-out phase.

We acknowledge that the flight trajectory reconstruction might not be strictly valid. However, it serves the main objective of this study by enabling the simulation of a short interval that encompasses a figure-of-eight manoeuvre during reel-out. A more educated reconstruction would require a lot more resources and probably more testing and is recommended as a possible future improvement.

## Appendix B: Pitch and roll angle definitions

Expressing the attitude of the kite using pitch and roll angles with respect to the wind reference frame gives large variations of these angles along the flight trajectory. Consequently, the kite attitude is difficult to interpret from these angles. Variations are smaller when the pitch and roll angles are expressed with respect to the tangential plane, which is perpendicular to the position vector of the kite and shown with the black rectangle in Fig. B1. The variations are smaller since the up-direction (positive z-axis) of the kite and the direction of the position vector in the wind reference frame are not far apart, especially during the

reel-out phase, where the tether is relatively straight due to the high pulling force of the kite.

### B1   Measured attitude of the kite

The rotation matrix for the transformation from the earth to the tangential reference frame is calculated by:

$$\mathbb{T}_{\tau e} = \begin{bmatrix} \sin\hat{\beta} & 0 & -\cos\hat{\beta} \\ 0 & 1 & 0 \\ \cos\hat{\beta} & 0 & \sin\hat{\beta} \end{bmatrix} \begin{bmatrix} \cos(\hat{\varphi}+\hat{\varphi}_{\mathrm{we}}) & \sin(\hat{\varphi}+\hat{\varphi}_{\mathrm{we}}) & 0 \\ -\sin(\hat{\varphi}+\hat{\varphi}_{\mathrm{we}}) & \cos(\hat{\varphi}+\hat{\varphi}_{\mathrm{we}}) & 0 \\ 0 & 0 & 1 \end{bmatrix}, \tag{B1}$$

in which subscripts $\tau$, w, and e refer to the tangential, wind, and earth reference frames, respectively, the hat denotes a measured

quantity, $\beta$ is the elevation angle, and $\varphi$ is the azimuth angle.

    The measured pitch, roll, and yaw of the wing of the kite are expressed using 3-2-1 Euler angles. The corresponding rotation matrix for the transformation from the earth to the top wing surface reference frame is calculated by:

$$\mathbb{T}_{\mathrm{tws\text{-}e}} = \begin{bmatrix} 1 & 0 & 0 \\ 0 & \cos\hat{\phi} & \sin\hat{\phi} \\ 0 & -\sin\hat{\phi} & \cos\hat{\phi} \end{bmatrix} \begin{bmatrix} \cos\hat{\theta} & 0 & -\sin\hat{\theta} \\ 0 & 1 & 0 \\ \sin\hat{\theta} & 0 & \cos\hat{\theta} \end{bmatrix} \begin{bmatrix} \cos\hat{\psi} & \sin\hat{\psi} & 0 \\ -\sin\hat{\psi} & \cos\hat{\psi} & 0 \\ 0 & 0 & 1 \end{bmatrix}, \tag{B2}$$

in which subscripts tws and e refer to the top wing surface and earth reference frames, respectively, $\phi$ is the roll angle, $\theta$ is the

pitch angle, and $\psi$ is the yaw angle.

    The attitude of the kite is not affected by the depower signal and can be approximated by pitching the wing reference frame with the negative of the depower angle $\alpha_{\mathrm{d}}$ depicted in Fig. 3

$$\mathbb{T}_{\mathrm{b\text{-}tws}} = \begin{bmatrix} \cos\alpha_{\mathrm{d}} & 0 & \sin\alpha_{\mathrm{d}} \\ 0 & 1 & 0 \\ -\sin\alpha_{\mathrm{d}} & 0 & \cos\alpha_{\mathrm{d}} \end{bmatrix}, \tag{B3}$$

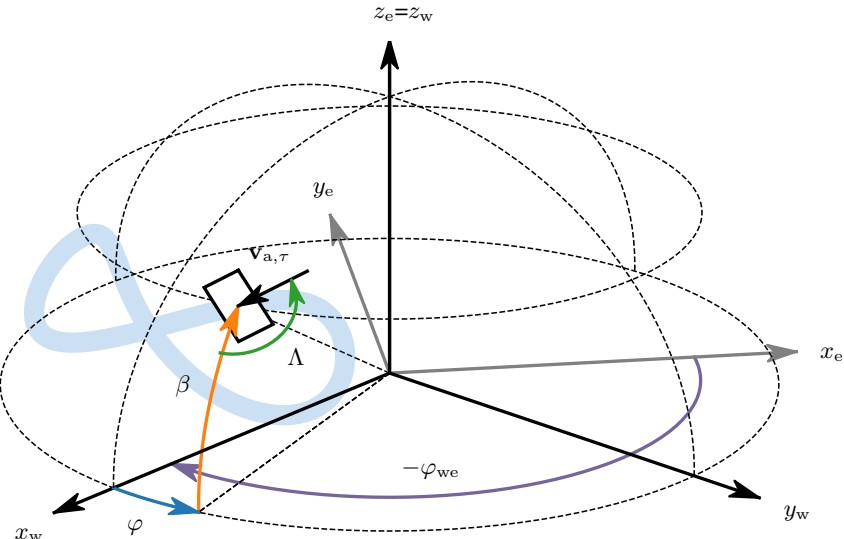

**Figure B1.** Earth reference frame $x_e, y_e, z_e$ and wind reference frame $x_w, y_w, z_w$ together with the yawed tangential plane lying on the projection of a figure-of-eight flight path. This plane is yawed such that it heads into the apparent wind velocity and serves as a departure point for expressing the kite attitude, illustrated in Fig. B2. The corresponding yaw angle $\Lambda$ is equal to the kite heading in case of zero side slip.

in which subscript b denotes the bridle reference frame. The depower angle is calculated using a geometrical model from the power setting (Schelbergen and Schmehl, 2020) and yields a nose-down pitch angle of roughly $6.6°$ during the reel-in phase.

The rotation matrix for the transformation from the tangential to the bridle reference frame is derived from the previously presented matrices:

$$\mathbb{T}_{b\tau} = \mathbb{T}_{b\text{-tws}}\,\mathbb{T}_{\text{tws-e}}\,\mathbb{T}_{\tau e}^\top. \tag{B4}$$

A rotation matrix can be represented with a set of 3-2-1 Euler angles. The yaw, pitch, and roll corresponding to these three angles can be calculated using the lower expressions:

$$\psi = \arctan2\left(\mathbb{T}_{12}, \mathbb{T}_{11}\right), \tag{B5}$$

$$\theta = -\arctan2\left(\mathbb{T}_{13}, \sqrt{\mathbb{T}_{23}^2 + \mathbb{T}_{33}^2}\right), \tag{B6}$$

$$\phi = \arctan2\left(\mathbb{T}_{23}, \mathbb{T}_{33}\right), \tag{B7}$$

in which $\mathbb{T}_{ij}$ denotes the transformation matrix element at the $i^{\text{th}}$ row and $j^{\text{th}}$ column. The Euler angles corresponding to $\mathbb{T}_{b\tau}$ are denoted without a subscript. The definitions of the pitch and roll angles are illustrated in Fig. B2, taking the yawed tangential plane as the point of departure.

$\Lambda$ in Fig. B1 describes the orientation of the tangential projection of the modelled apparent wind velocity, also shown in Fig. 5. In case of no side slip, $\Lambda$ equals the heading angle. The heading angle inferred from measurements and $\Lambda$ has a small

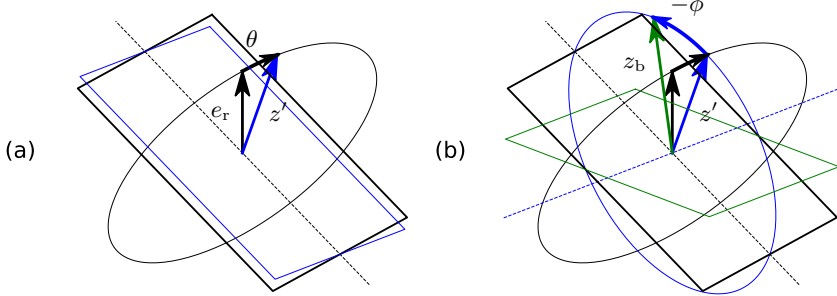

**Figure B2.** Last two rotations in the 3-2-1 sequence (Euler angles) to get from the tangential to the bridle reference frame: (**a**) a positive pitch rotation and (**b**) a negative roll rotation. The black rectangle illustrates the yawed tangential plane, introduced in Fig. B1.

periodic misalignment (not plotted), which may indicate a side slip. However, the constant wind assumption and measurement errors introduce too much uncertainty to confirm this. Also, the side slip angle was not measured in the studied test flight and thus can not be validated. Nevertheless, some side slip can be expected, as previously shown in the experiments by Oehler and Schmehl (2019).

## B2   Modelled attitude of the kite

Expressing the Euler angles of the kite element of the model requires assigning a local reference frame to the element. The model does not specify a full reference frame but only specifies the axial direction of the element. This axial direction is used as the z-axis for the local reference frame. To differentiate between the roll and pitch, also the x-axis and y-axis need to be specified. The x-axis is chosen such that it lies in the plane spanned by the position vector and the vertical direction $z_e$. The y-axis then follows from the other two axes and is oriented horizontally.

Other than for securing the alignment between the roll and pitch definitions of the measured and modelled kite attitude, the yaw of the tether is not of interest to this study. It does not affect the kite attitude itself, and therefore, the resulting yaw angles are left out of Fig. 11. The modelled yaw of the kite is similar to that inferred from the wing attitude measurements and, thereby, facilitates comparing the measured and modelled roll and pitch.

*Author contributions.* Conceptualisation, M.S. and R.S.; methodology, M.S.; software, M.S.; investigation, M.S.; writing—original draft preparation, M.S.; writing—review and editing, R.S.; supervision, R.S.; funding acquisition, R.S. All authors have read and agreed to the published version of the manuscript.

*Competing interests.* Roland Schmehl is a member of the editorial board of Wind Energy Science. He is also a co-founder of and advisor for the start-up company Kitepower B.V., which is commercially developing a 100 kW kite power system and provided their test facilities and staff for performing the in situ measurements described in this article. Both authors were financially supported by the European Union's Horizon 2020 project REACH, which also provided funding for Kitepower B.V.

*Acknowledgements.* This research was part of the project REACH (H2020-FTIPilot-691173), funded by the European Union's Horizon 2020 research and innovation programme under grant agreement No. 691173, and the project AWESCO (H2020-ITN-642682), funded by the European Union's Horizon 2020 research and innovation programme under the Marie Skłodowska-Curie grant agreement No. 642682. The authors are grateful to Kitepower B.V. for making the flight data available in open access and for sharing expertise about the system, in particular, Joep Breuer for asking critical questions. Also, we would like to thank Arthur Roullier whose MSc thesis this work builds upon

and Jochem de Schutter for his tips on implementing the dynamic simulation.

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
