# Peer review of "Swinging motion of a kite with suspended control unit flying turning manoeuvres"

_Wind Energy Science, 2023_

## Referee Comment (RC1)

The submitted paper is relevant to the airborne wind energy community and addresses a modelling aspect of the kite dynamics that has, to the knowledge of the reviewer, not been addressed yet. The contribution of the paper is well-defined, and the scientific approach and methodology are clearly described. The paper is well written.

The reviewer does not note any flaws in the presented results, interpretations, and conclusions. However, some mention of Kitepower BV may be not necessary for the scientific purpose of the paper. The main comments are regarding some clarification that could be made and some advice to ease the reader's understanding. The list of comments is listed below.

5 : and ... and

11 (and 414) : "Hence, intricate centripetal force modelling is avoided, as seen in a single-point kite model." Could you provide a reference where this is done/needed or explain this in more detail. You also mention it in line 47, maybe you can elaborate on the need for the modelling of this force and give an example somewhere (Section 5 ?), for the one-point model.

19 : Figure 1 is a nice picture but I do not think it is the best to illustrate the hanging KCU when you then show Figure 3.

77 : Why are we talking about the 60 m2 kite of Kitepower as it is not the subject of the measurement campaign ? If it is for advertising, it has no place in a scientific article.

119 : You should cite a reference or give the classical assumptions made for the one-point model compared to the one-point model.

191 : maybe refer to Appendix A for the optimization ? Or mention the optimization when citing or introducing your Appendix earlier.

251 : You previously often made mention of the point model and stated the advantages of the 2-point model against this last one. Would it be possible to compute the results with a one-point mass model at the centre of gravity of the kite + KCU, with 30 elements, for example?

268 : Tangential to what ? (sphere on which the kite is navigating I guess)

292 : The description of Fig 11 is not consistent with what is explained "The measured force shows distinct peaks during the turns" while the large peaks are from the blue line, identified as from "Dynamic model"

313 : Where is the comparison shown? Fig 10 ? Maybe say it already here

Fig 12 is not in a PDF format; any averaging operation for the orange curve ?

330 : add a comma "This discrepancy could be attributed to the high uncertainty of the position measurement during the turns, resulting in large modifications to the flight trajectory by the reconstruction. "

358 : Maybe specify which tip you are talking about. Interior or exterior tip for the increase/decrease in aoa.

365 : "Per definition, the lift force generated by the kite rolls together with the kite and, when rolled, contributes to the centripetal force acting on the KCU " This sentence is not very clear.

Appendices :
Fig A1: it is not very clear what is plotted, nor explicitly explained; Fig not in pdf format
The discussion of the Figure is a bit confusing. The discussions are not made in the order a, b, c, … you talk about A1c, then A1a, then the other, then A1c again. In my opinion, the structure of this appendix could be improved to ease the understanding. In addition, some of the symbols are not described (what is $v_{c,r}$ ?).

---

## Referee Comment (RC2)

[revised manuscript text omitted]
{s,r}} = \frac{\mathbf{r}_{\text{s}}^T \mathbf{v}_{\text{s}}}{\|\mathbf{r}_{\text{s}}\|}, \tag{A1}$$

435 in which $\mathbf{r}_{\text{s}}$ and $\mathbf{v}_{\text{s}}$ are the position and velocity of the wing, respectively. The flight trajectory reconstruction needs to assure consistency between the kinematic properties such that also the updated radial component of the wing velocity and the derivative of the radial position will agree.

As an additional check, the derivative of the radial position of the wing is compared to the reel-out speed. The derivative of the radial position shows large fluctuations around the tether reel-out speed in the reel-out phase. The magnitude of the
440 fluctuations conflicts with our expectation that the changes in tether slack and stretch are small in this phase. Towards the end of the turns, the derivative of the radial position even tends to become shortly negative, which coincides with the subtle local maxima found at the end of the blue intervals in the recorded radial position in Figure A1a.

Figure A1a shows how the integrated measured reel-out speed (dotted black line) evolves with respect to the recorded radial position of the wing (blue line). During the right turns, the inferred tether length stays increasing roughly linearly, while the
445 radial position shows local maxima. Note that the lines need to be shifted up with their initial values to obtain their respective absolute values. Unfortunately, we do not know the absolute tether length as it is not measured directly.

The residual between the inferred tether length and recorded radial position is shown in Figure A1b. During the right turns, the residual changes roughly 2 m (depth of the valley) within a couple of seconds. The corresponding relatively large increase in radial position can partly be attributed to a decrease in tether slack and an increase in tether stretch. However, the magnitude
450 of the change is deemed to be too large to be attributed only to tether dynamics. Note that also here, the line may shift vertically depending on the initial values. As such, we can not draw conclusions based on the magnitude of the residual but merely on how it changes with time. The sum of the residual length and an unknown offset gives an approximation of the tether slack. The true tether slack also accounts for tether stretch, which is not accounted for in calculating the residual length. Note that the tether slack can not be negative.

455 The maxima in the recorded radial position do not need to be purely physical. Another possible cause is GPS inaccuracy during manoeuvres, which has previously been reported in the literature. Borobia et al. (2018) reported measured radial position exceeding varying more than 3 m while none was expected. Considering the imprecision of the recorded position, we opt for

[Figure]

[Figure]

**Figure A1.** (**a**) Evolution of unstrained tether length and the recorded and reconstructed radial distances of the wing, all with their initial values subtracted. (**b**) Difference between the tether length and radial distance. (**c**) Measured tether reel-out speed and recorded and reconstructed radial speeds of the wing. (**d**) Residual between the reel-out speed and reconstructed radial speed. The intervals shaded grey and blue indicate left and right turns, respectively, from a downwind perspective.

adapting the wing kinematics and letting the tether reel-out acceleration input follow the measured reel-out speed as closely as possible.

[Figure]

[Figure]

The flight trajectory reconstruction is obtained using an optimal control problem that minimises the error between the
modelled radial wing speed and recorded tether reel-out speed while limiting the bias between the modelled and recorded wing
position

$$\min_{\mathbf{r}_\mathrm{s}, \mathbf{v}_\mathrm{s}, \mathbf{a}_\mathrm{s}, r_\mathrm{s,r}, v_\mathrm{s,r}, a_\mathrm{s,r}} \quad \sum_{i=0}^{N} \left[ \mathrm{w} \left( v_\mathrm{s,r} - \hat{\dot{l}}_\mathrm{t} \right)^2 + (\mathbf{r}_\mathrm{s} - \hat{\mathbf{r}}_\mathrm{s})^T (\mathbf{r}_\mathrm{s} - \hat{\mathbf{r}}_\mathrm{s}) \right]_{t=\frac{i}{10}}$$
$$\text{s.t.} \quad \mathbf{a}_\mathrm{s} = \dot{\mathbf{v}}_\mathrm{s} = \ddot{\mathbf{r}}_\mathrm{s} \qquad\qquad\qquad ;$$
$$a_\mathrm{s,r} = \dot{v}_\mathrm{s,r} = \ddot{r}_\mathrm{s,r}$$
$$r_\mathrm{s,r} = \|\mathbf{r}_\mathrm{s}\|$$
(A2)

[revised manuscript text omitted]

---

## Author Response (AR1)

Dear **reviewer #1**,

Thank you for the comprehensive review of our manuscript. In the following, we will reply to your comments point by point.

The submitted paper is relevant to the airborne wind energy community and addresses a modelling aspect of the kite dynamics that has, to the knowledge of the reviewer, not been addressed yet. The contribution of the paper is well-defined, and the scientific approach and methodology are clearly described. The paper is well written. The reviewer does not note any flaws in the presented results, interpretations, and conclusions. However, some mention of Kitepower BV may be not necessary for the scientific purpose of the paper. The main comments are regarding some clarification that could be made and some advice to ease the reader's understanding. The list of comments is listed below.

5: and … and

Response: rephrased sentence

11 (and 414): "Hence, intricate centripetal force modelling is avoided, as seen in a singlepoint kite model." Could you provide a reference where this is done/needed or explain this in more detail. You also mention it in line 47, maybe you can elaborate on the need for the modelling of this force and give an example somewhere (Section 5 ?), for the one-point model.

Response: The intricate centripetal force is a conclusion based on the literature review. Therefore, I expanded the paragraph in the introduction, which covers the work of Fechner, to give an example of intricate centripetal force modelling and explain that it originates from not resolving the roll. It should be more clear now.

19: Figure 1 is a nice picture but I do not think it is the best to illustrate the hanging KCU when you then show Figure 3.

Response: We agree that there are probably better photos to depict the suspended kite control unit. However, this photo is one of the rare available ones that shows the entire system (kite, KCU, tether, and ground station), which, we think, is a useful contribution to the introduction. The photo in Figure 2 and the drawing in Figure 3 focus on the kite, and from these, the configuration should be clear.

77: Why are we talking about the 60 m2 kite of Kitepower as it is not the subject of the measurement campaign? If it is for advertising, it has no place in a scientific article.

Response: The purpose of this photo is to show the commercial derivative of the system in a relevant environment. We think that this photo is quite well suited for this purpose, indicating the actual application of the technology, fitting well in the introduction. We removed the manufacturer's name from the actual caption (leaving it only for the courtesy statement in the brackets).

119: You should cite a reference or give the classical assumptions made for the one-point model compared to the one-point model.

Response: On hindsight, discussing the one-point model here seems distracting. Therefore, these first two sentences have been removed.

191: maybe refer to Appendix A for the optimization ? Or mention the optimization when citing or introducing your Appendix earlier.

Response: The optimisation problem is added directly to the text as it is not directly related to Appendix A.

251: You previously often made mention of the point model and stated the advantages of the 2-point model against this last one. Would it be possible to compute the results with a one-point mass model at the centre of gravity of the kite + KCU, with 30 elements, for example?

Response: Analysing the roll and pitch of the kite under an imposed flight path is the main topic of this paper. The 1-point model does not resolve these and thus is irrelevant to this section. It would have been relevant if the full motion had been resolved, i.e., if the flight path of the wing was not constrained. The statements about the 1-point model purely arise from the literature review.

268: Tangential to what? (sphere on which the kite is navigating I guess)

Response: added "(perpendicular to the position vector of the kite)"

292: The description of Fig 11 is not consistent with what is explained "The measured force shows distinct peaks during the turns" while the large peaks are from the blue line, identified as from "Dynamic model"

Response: corrected

313: Where is the comparison shown? Fig 10 ? Maybe say it already here

Response: Included reference to figure.

Fig 12 is not in a PDF format; any averaging operation for the orange curve?

Response: Changed to PDF. The caption of fig 12 mentions how the orange lines are generated: "The orange dashed lines in the left column depict the steering input scaled with the slope found in the linear fit shown with the orange dashed lines in the right column."

330: add a comma "This discrepancy could be attributed to the high uncertainty of the position measurement during the turns, resulting in large modifications to the flight trajectory by the reconstruction. "

Response: Comma added

358: Maybe specify which tip you are talking about. Interior or exterior tip for the increase/decrease in aoa.

Response: Incorporated suggestion

365: "Per definition, the lift force generated by the kite rolls together with the kite and, when rolled, contributes to the centripetal force acting on the KCU" This sentence is not very clear.

Response: Removed "Per definition"

Appendices:
Fig A1: it is not very clear what is plotted, nor explicitly explained; Fig not in pdf format The discussion of the Figure is a bit confusing. The discussions are not made in the order a, b, c, … you talk about A1c, then A1a, then the other, then A1c again. In my opinion, the structure of this appendix could be improved to ease the understanding. In addition, some of the symbols are not described (what is $v_{c,r}$ ?).

Response: Fig changed to pdf. In the figures, we chose to represent the quantities in the order of lengths first and subsequently the time derivatives, or speeds. However, this is not per se the most logical order to discuss the results. It was chosen to first discuss the speeds because this displays better what the problem is. We improved the text with more explicit references to the lines in the figures.

Dear **reviewer #2**,

Thank you for the comprehensive review of our manuscript. In the following, we will reply to your comments point by point.

General comments

In the manuscript "Swinging Motion of a Kite with Suspended Control Unit Flying Turning Manoeuvres", the authors present an interesting and insightful analysis of the physical mechanism inducing the turns in kites with suspended control units. The manuscript is well-structured and clearly organized.

In my opinion, the main outcome of this work is the physical understanding of the mechanisms inducing the turns. However, this message is not given in the abstract and could be better clarified in some parts of the text. I suggest giving more importance to this result.

Response: We think that the comparison of the modelled and measured swinging motion is the core contribution of the paper. The motion analysis strongly suggests that the roll of the kite is a key enabler of the turning. However, we also think that additional scientific evidence is needed to firmly support the proposed turning mechanism and to present it as the main scientific contribution of a paper.

I have attached a pdf with my comments and I summarize here the main technical comments that should be addressed.

1. The aerodynamic drag is generated only by the component of airflow perpendicular to the segment. This means that a component of the velocity parallel to the tether element would not generate any aerodynamic force. This should be corrected in eq. 5.

Response: changed as suggested

2. The assumptions and mathematical derivations in Section 3.2 should be clarified. Is there a rigorous physical derivation for omega_turn? It would be nice if the straight omega and the turn omega could be derived as particular cases of a generic omega.

Reponse: Thank you for this suggestion. After some more thought, a concise expression was derived.

3. The models with one single tether element for the tether from the ground-station to KCU and one element from the KCU to the kite employ a lumped drag model (eq. 7) and not the classic finite element method (eq. 5). I believe that calling this case "single tether element" is misleading, as the lumped drag model can be derived from continuous models of the tether (with infinite tether elements) under prescribed assumptions. I would name this case differently.

Response:  With single tether element, the paper refers to how the tether is discretized, not to what the lumping approach is used. The lumping approach can be chosen freely, independently of the discretization approach. To summarize this:
   1. multiple tether element -> eq. 5 is used for the drag evaluation;
   2. single tether element -> eq. 5 is used for the drag evaluation;
   3. lumped tether drag (or different naming) -> eq. 8 is used for the drag evaluation.

2. Comments in pdf

Page 1:

I would reduce this part of the abstract and add a couple of sentences about the physical mechanics driving the turn, which to my opinion is the main contribution of this work and it is not mentioned here.

Response: As mentioned above, we think that the comparison of the modelled and measured swinging motion is the core contribution of the paper. The motion analysis strongly suggests that the roll of the kite is a key enabler of the turning. However, we also think that additional scientific evidence is needed to firmly support the proposed turning mechanism and to present it as the main scientific contribution of a paper.

However, in many papers, this mechanism is already incorporated to drive turns, albeit less so for flexible kite systems.

Please add the references, when they are not part of the sentence, between round brackets "()"

Response: fixed

Page 3:

Giving this data (especially without introducing the side force coefficient) is in contrast with the qualitative discussion given in the introduction.

Same comment as for the side force coefficient.

Response: The paragraph is rewritten and the reason for mentioning this values is clarified; the lateral force coefficient of 2.59 in the single-point kite model is very high compared to the lift force coefficient of the wing tip in the five-point kite model, suggesting that the single-point model is not using a realistic turning mechanism.

Page 5:

Is there any specific reason to take this reference system with the y and z axis opposite to what typically consider in flight mechanics? (just out of curiosity, it is not necessary to modify it as the reference systems can be set arbitrarily ).

Response: There is no specific reason. We used this convention in line with several performance models developed by the TU Delft research group.

Page 6:

can you make this line thicker?

Response: done

not clear what default stands for here

Response: Sentence expanded: The default Kalman filter implementation of the Pixhawk was used to enhance the quality of the measured kinematic quantities.

Page 7:

bold italic should be used for vectors, while bold for matrices.

Response: We follow the "Reviews of Modern Physics" style guide in which vectors are typeset in bold roman (upright) font. See https://cdn.journals.aps.org/files/rmpguide.pdf. We do acknowledge that there are also other style recommendations, as the one by IUPAC, suggesting the use of bold italic.

Not sure what you mean here. With "align" you mean "parallel"? This would not be true for an AWES. I would anyway rephrase the paragraph, removing this sentence as it is not necessary, to highlight the key message in the last sentence.

Response: Could you say that two lines align? [ChatGPT] Yes, you can say that two lines align. This phrase is commonly used to indicate that the two lines are in a straight and coincident position, meaning they share the same trajectory or direction. It suggests that the lines are parallel or perfectly matched in some way. [MS] Rewritten sentence: "The side slip angle is the angle between the heading of the kite and the apparent wind velocity."

"dynamic equations of motion of the kite"? not clear what "unconstrained" means here

Response: Not clear what you mean with the first sentence. But we removed this sentence as it was a bit confusing.

Page 8:

A reference system attached to the kite would help understanding around which axis the yaw, pitch and roll are measured and in which sense.

Response: I was hoping to avoid introducing another reference frame. Especially if it is not used in the calculations. If you think the current figure is really misleading, I can also remove the arrows indicating yaw, pitch and roll from the figure.

Pay attention here: for the crossflow principle, the aerodynamic drag is generated only by the component of airflow perpendicular to the segment. This means that a component of the velocity parallel to l_j would not generate any aerodynamic force. This expression does not capture this phenomenon.
Zanon et al. investigate Fly-Gen AWESs and consider as velocity in this expression the velocity at the mid point of the segment. For Fly-Gen that expression is a good approximation, as the tether is not reeled out and the apparent wind speed perpendicular to the segment is similar to its motion. For Ground-Gen this is slightly different, as the reel-out is actually along the tether direction and neglecting this effect might lead to an unphysical force along the tether direction. I don't expect results to change significantly, but please check on this and on the following expression for the drag forces. See eq. (6) in Williams (2017) for the correct formulation.

Response: I chose to change it to calculate drag only with the normal component as suggested. However, only in the Williams' model and not in the dynamic model. It affected the results slightly but the conclusions remain unchanged.

.. on the KCU point mass

Response: rephrased

Page 9:

It would be nice to have a comment on how this coefficient is estimated

Response: Choosing the value was a bit arbitrary. Added to the text: "The chosen value of 1.0 for the drag coefficient is within the common range for a blunt body."

I got a bit lost in this section.
Why are we looking for an angular velocity representative of the turn? Williams' formulation does not work for turn maneuvers? It seems to me that it does (the left-hand side of Eq. 17 of Williams computes centrifugal forces, or am I wrong?).
If it works for turns, I would formulate Eq. 10 in a general way (i.e. without the pedix "straight"). Then, I would state that we look for an expression for the straight part and for the turn. The formulation for the straight part is equal to Williams formulation, while the formulation for the turn is approximated with eq. 11.

Response: I do not consider following a great-circle path along the surface of a sphere with fixed radius as turning. However, with such flight path there are still centrifugal forces acting on the system, which are considered by Williams. Explanation added: "This rotational velocity is labelled as 'straight', because the great-circle rotation produces the straight path segments depicted in Figure 4."

or more generally: "inertial force"? so that also Coriolis is included.

Response: Reformulated: "This force balance considers.." instead of "consists".

I think that the fact that Williams' formulation does not capture inertial forces (and hereby you need to define two omegas) comes from the assumptions of:
1) rigid body
2) null angular acceleration
I would treat the derivation in a more general way, without these assumptions, and adding them, if needed, later.
What if you model the a_j in eq. 9 (which is the same as the left-hand side of eq. 17 in Williams) not by writing eq. 14 to 16 (Williams), which use assumptions 1 and 2, but by considering the actual node acceleration?
In spherical coordinates, the node acceleration would be (https://ocw.mit.edu/courses/16-07-dynamics-fall-2009/57081b546fff23e6b88dbac0ab859c7d_MIT16_07F09_Lec05.pdf)
$a = (\ddot{r} - r\dot{\theta}^2 \cos^2 \varphi - r\dot{\varphi}^2) \, e_r$
$(2\dot{r}\dot{\theta} \cos \varphi + r\ddot{\theta}\cos \varphi - 2r\dot{\theta}\dot{\varphi}\sin \varphi) \, e_\theta$
$(2\dot{r}\dot{\varphi} + r\dot{\varphi}^2 \sin \varphi \cos \varphi + r\ddot{\varphi}) \, e_\varphi$ .

where you should have everything, if I am not misunderstanding. If you want to use it as a quasi-steady model, then you could drop the second derivatives of r, phi, theta, but keep the other inertial term.
Can you then relate the omega_straight and omega_turn to this?
If not, could you give a bit more context to this? Maybe by adding the key equations from Williams. This would help following the discussion without having to open Williams paper.

Response: Williams considers one inertial force contribution which is the centrifugal force. The inertial forces experienced by an observer (or the kite) in a rotating refence frame are Euler force (-mw'xr), Coriolis force (-2m(wxv)), and centrifugal force (-mwx(wxr)) - this does not become more clear if you write it out in spherical coordinates in my opinion. We assume no rotational acceleration so Euler force is zero and no velocity of the kite wrt the rotating ref. frame (we actually infer the angular velocity based on the velocity of the kite) so Coriolis force is zero.

I rewrote the introduction of the equations a bit: "To facilitate the calculation of loads, the velocities and accelerations of the point masses are approximated by assuming that they collectively rotate around the tether attachment point at the ground with a constant angular velocity, treating the point masses as particles of a rigid body. Consequently, the velocity and acceleration of each point mass depend solely on the angular velocity and its respective position."

Page 10:

please, move the cross product at the numerator, to be consistent with eq. 10

Response: done

with respect to the model from Williams?

Response: added

Page 11:

This should be the other way around, as discussed earlier.

Response: changed as discussed earlier

Page 12:

why "c" for wing? I assume because "w" is already taken from wind, but then what "c" stands for?

Response: this was a bit confusing indeed, changed to "k" for kite.

Page 13:

This is a really interesting method.

Response: :)

If you use eq. 7 for the drag of the first element, you are actually lumping the drag acting on the first element with the KCU. Eq. 7 cannot however be used for the second element. How do you split

the drag of the second element between the two point masses? Do you use eq. 5 with half of the second element length on the KCU point mass and half on the kite mass? I think this would be reasonable.

Response: This would be a good option, however it is not exactly how it is implemented. In the minimal model (two points), the second segment represents the kite for which the bridles produce drag. However, here we didn not include the bridle drag as separate term as mentioned in 3.1: "The bridle and ram-air turbine drag are not included as separate terms but are considered accounted for by the KCU and wing drag."

Page 15:

It should not contribute for the cross-flow principle, as discussed earlier.

Response: Sentence removed.

The velocity triangle and thus the apparent wind direction changes according to the tether position. Moreover, the drag is not actually a function of the tether's apparent wind velocity, as discussed earlier.
Wouldn't it be better to plot the tether shape with the kite motion direction and cross-motion?

Response: Indeed, the caption is clarified to better explain what is done: "Tether-kite lines with cross-axial displacement decomposed with respect to the tangential apparent wind velocity of the wing (see Figure 4)." This decomposition was considered more suitable than the suggested decomposition because the contribution of drag is less prominent in the cross-direction. Thereby, the effect of other loads is easier to observe.

Page 17:

The single element tether drag in Eq. 7 can also be derived by solving the continuous displacement equation of the tether subject to exclusively aerodynamic drag in steady-state. See "The Influence of Tether Sag on Airborne Wind Energy Generation" (Trevisi et al.).
Therefore, I think that it can be misleading to refer to the evaluation of the drag with Eq. 7 as "single tether element": it is instead derived with infinite number of elements (continuous). I would suggest something like "lumped drag". A single tether element would still have the drag computed with eq. 5.
It would be interesting to check the tether slope at the KCU (eq. 3 in Trevisi) and eventually the maximum displacement obtained with this analytical model against the numerical results with 30 tether elements.

A similar discussion can be done for the tether mass. A single element tether would have the mass distributed as in eq. 4. A lumped model would have one third of the first element tether mass on the kcu when evaluating the centrifugal forces. This is derived in Sect 4 of Trevisi for circular trajectories. It might be not fully applicable to figure of eight.

Response: With single tether element I am refering to how the tether is discretized, not to what the lumping approach is used.

Page 21:

The side force is the component of the resultant aerodynamic force..

Response: Rewritten: "The introduction of a side component rolls the resultant aerodynamic force acting on the whole kite without rolling the kite itself." Hopefully this is more clear?

Page 22:

I would rephrase with something like: "the kite starts to roll to increase the centripetal aerodynamic force as it needs to pull the relatively heavy KCU into the turn"

Response: Rephrased to: "the kite starts to roll to exert a centripetal force on the relatively heavy KCU and pull it into the turn."

You said a few sentences earlier that the resultant aerodynamic force is not in the symmetry plane of the kite because of the wing twist.
Therefore, the mechanics imposing the centripetal force is that the direction of the resultant aerodynamic force point inward (and not directly the roll of the kite, which is a consequence of the twist and of the kcu pulling outwards). This sentence should be reformulated, if I am not misunderstanding.

Response: Reformulated: "The greater the roll of the kite relative to the upper tether end, the more the aerodynamic force of the top wing surface contributes to the centripetal force."

This paragraph is very important for the understanding of the whole paper and I think it lacks a bit in clarity. I would try to highlight the mechanisms which initiate and drive the turn by separating the 2 phenomena:
1) modification of the resultant aerodynamic force by twisting. This just modify the direction of the resultant aero force by creating a inward force in the kite reference system
2) roll motion of the kite, driven by the fact that link between kcu and kite is rigid, so that the kite motion is just allowed on a sphere centered on the kcu.

Response: Reordered these two paragraphs: 'topic' sentences are now at the start of paragraphs.

Dear **reviewer #3**,

Thank you for the comprehensive review of our manuscript. In the following, we will reply to your comments point by point.

My overall assessment:

=====================

The authors of this paper are trying to better understand how the mass of the control unit effects the dynamics of the a soft wing airborne device during turns. The study is based on some noisy data from experimental tests. The authors used some optimization to smooth out this experimental data to give estimates of the path, velocity and acceleration. The authors create 2 models to better understand these dynamics. The first model is a quasi-static model that ignores dynamic effects, while the second is a dynamic model that accounts for these transient effects. To allow the authors to ignore aerodynamic forces, they run both models on fixed paths based on the kinematic data extracted and smoothed from the test data. The study shows that during turns, the inertia of the control unit will induce a roll rotation on the wing, which in-turn provides the centripetal force needed to turn the control unit. The authors also make several comparisons between the two models and other measured data to help validate these models. The authors also state that these 2 mass models may be more appropriate for control design. It's my opinion that this is an interesting study and worthy piece of work for publication.

However, the presentation of this work can be greatly improved. Overall there are a lot of important details that are not well explained, vague or presented in ways that add to the confusion. I was very confused on many parts of the manuscript and it was difficult for me to judge the different results. I have made several comments below that outline all the different points where I think the presentation can be improved. Overall, the results show agreement between the models and the experimental data. However, there are also plots that show poor agreement. Furthermore, it's unclear how different assumptions, test conditions/configuration, modelling error, data post-processing error could have contributed to the agreements or errors in these models and the results. The author doesn't really give a satisfactory explanation of these points. So this is second point where I think the manuscript could be further improved.

Given the difficulty of working with noisy experimental data, and the added complexity of these two models, it's clear that this was likely a difficult study to carry out. So despite these problems in the presentation, I have a positive opinion on the overall quality of the scientific work. So I recommend that this work is published once the presentation is improved.

Further comments:

=================

The title could be improved

    First, you need a conjunction between Unit and Flying

  Idea 1: Swinging Motion of a Kite with Suspended Control Unit while Flying Turning Manoeuvres

Next, I am not sure you are talking about 'swinging motion', but more the interaction of this control unit and the wing in these turns so the next idea is as follows. 'Swinging' brings up thoughts of pendulum motion, but what you have here is different than that.

Idea 2: The dynamic interactions of a kite wing and control unit during turning manoeuvres

Response: Thank you for these ideas, which are both very useful alternatives to our original title. We have spent quite some iterations on the title and are still confident about it, as is. We started out with a much longer, convoluted title, until we found this one.
About your first point: we do not see the need for a conjunction as you pointed out. The title is a short form. We believe that the short from justifies leaving out a conjunction in "kite flying turning maneuvers", similar to "aircraft flying in formation" which we find in many photo captions in the public literature. The addition "with Suspended Control Unit" does not change this, in our opinion, and there are many examples of similar titles in the literature. Adding a "while" is optional, but we would like to keep the title as brief as possible.
About your second point: we do find the mentioning of the 'swinging motion' very illustrative. The back-and-forth lateral motion of the KCU during the turns (induced by the inertia of the KCU mass) is a characteristic motion pattern of this type of kite, which resembles swinging. One could argue that the relative motion of the KCU with respect to the top of the kite mimics the swinging motion of a pendulum beneath its anchoring point, even though the kite's top itself is actually moving. "Dynamic interaction" would also fit here, but this is less concretely describing this motion pattern. To conclude, we would like to adhere to the present title. We did make some changes to the text to emphasize that the swinging motion pertains to the motion of the KCU relative to the top of the kite.

Line 3: how accurately => the accuracy

Response: Sentence rewritten

Line 5: Inserted a 'be' " The motion of the wing point mass is constrained to be a figure-of-eight manoeuvre from the flight data of an existing prototype"

Response: Changed as suggested

In my opinion the abstract is much more detailed and technical than it should be. This effects the readability in my opinion. I needed to read it multiple time to understand what it was saying. I think this paper could benefit from having the abstract written to explain the study and the results with higher level language. This would make it much more attractive for people browsing the literature.

Response: Agree, abstract rewritten.

In the introduction, the author is explaining the configuration of the kite, how the wing deforms, a range of models according to their kinematic description. It could be helpful to have a diagram showing these different descriptions to help make sense of it.

Response: We believe that there are already a lot of figure as is. Therefore, we chose not to add a diagram.

Line 62: The wording of the hypothesis is vague and I am confused by the hypothesis. First you posit that the roll (I presume this is rotation, not a spelling mistake like role) is induced by the

inertial of this KCU. It's the second part of the sentence that is vague and confusing, it's a rather vague statement to say "... has a crucial role in the turns". So first, is it the roll or the inertia that has this crucial role? What is this crucial role? The turning force? the radius of the turn? generating the aerodynamic force? The time delay between actuation and effect? Is this link between the KCU inertia and it's effect on roll in question? (in other words, is this also a research question?)

Response: Good point. The existence of a relationship between KCU inertia and kite roll is quite evident. The research question is rather:
Which dynamics (i.e. centrifugal forces, tether deformation, transient effects) make a significant contribution to the observed characteristic pitch and roll swinging motion during sharp turning manoeuvres?
Sentence rewritten: "The goal of this paper is twofold: to study the dynamics of the observed characteristic pitch and roll swinging motion during sharp turning manoeuvres and discuss the implications to performance modelling."

In my opinion this paper has multiple contributions that aren't always stated so clearly in the abstract and the introduction. First it uses experimental data and two different models to explores the dynamic interaction between this control unit, the wing and the tether. Second it introduces two different models for modelling this behaviour. The paper does suggest in multiple locations that such a model could be an improvement in control design over 1 mass models. So the second contribution is these two models and some effort in trying to validate these models. So I would try to be more clear in stating these contributions.

Response: Paragraph in introduction is rewritten according to the contributions according to authors:
1. Presenting the 2-point kite model.
2. Validating the swinging motion computed with a 2-point kite model with both a straight and discretized tether and the motion approximated as a transition through steady-rotation states.
3. Exploring the impact of transient effects on the modelled swinging motion.
4. Investigation of the mechanisms that initiate and drive a turn in a flexible kite system with a suspended control unit.
(5. Identification of the correlation between the steering input and the local pitch of the wing at the inner struts.)

Line 95, the position of these pixhawk sensors could be labeled in figure 2, 3 or 5 just so that it is clear

Response: Added 'Pixhawk' to the labels in figure 3

Figure 4: This work depends heavily on the kinematic information of specific points of the wing and control pod. It would be good if the author made a more clear statement about what point on the kite figure 4 describes. I guess that it is the control pod?

Response: Reorderd text such that position measurement is moved forward and added 'of the wing' when describing the figure of eight.

Figure 5, the so-called wind reference frame is described in the caption of figure 5, the same frame is used in figure 6. In figure 5, it could be useful to state that y is in the direction of the wind. Furthermore, in figure 6, just restate that the coordinate system is the same as the one described in figure 5.

Response: It is the x_w-axis that is aligned with the wind velocity vector. We have added this information.

I think figure A1 needs to be moved to section 2. Since the study depends heavily on the kinematic data, it's important that this data is shown in section 2. I can understand that it's better to keep most details in the reconstruction in the appendix, but again, due to the importance of this, I would state in section 2, that the this data was solved with an optimization.

Response: I introduced the optimization textually in section 2 and metioned the limited validity. Also added a new figure with the outcome of the reconstruction.

Section 2: As stated earlier, this work depends heavily on the kinematic information of specific points. So I would like a more clear description of where the sensors were located, what sensors were used to generate what kinematic data. Since not all sensors were co-located, how did you assume that measurements from one location were related to another location? Did you assume the kite and bridle were rigid?

Response: The text states that the Pixhawk's were mounted to the wing at the inner struts. This is made more explicit in Figure 3. No other position or attitude measurements are conducted. The position of the control unit is only computed.

Line 199 "We allow for rotations other than great-circle rotations", maybe a mistake, but I assume 'we allow for no other rotations ...''

Response: No mistake, we do allow for rotations other than great-circle rotations (orange in figure 6). The other considered rotation is the turn circle in blue.

I have a lot of things to say about the description of the dynamic model in section 3.3:

- It's difficult to fully understand equation 12, first a diagram of the kinematic set-up could be helpful.

Response: added reference to figure 5. "For brevity, we present only the model formulation with a single tether element. This tether element directly connects the point mass of the control unit $m'_\mr{kcu}$ to the ground station, as opposed to using five tether segments as shown in Figure 5."

- I get that you are only looking at 2 point masses. Then you can say, row 1 is the equations of motion for the wing and row two for the tether connection point (maybe I am wrong, but there is some confusion with the subscripts).

Response: subscripts fixed. Text changed: "The differential equations in the upper two rows of Eq.~\ref{eq5} consist of the equations of motion of the control unit and the wing. The algebraic equations in the lower two rows represent the links between the point masses."

- Furthermore, your explaination of the different sub-scripts are difficult to understand because you have kcu, and s which are not explained. It would be better to have less subscripts and to make them consistent.

Response: added explanation of kcu. Subscripts are made consistent throughout paper.

- The bottom two rows, where you enforce your constraints, you should show your 'r' terms as transpose to indicate that it's a dot-product with the acceleration.

Response: corrected

- Second, these constraint equations are the derivative of equations 14, and 16 ... basically you are constraining the acceleration to be consistent with the enforced length. Enforcing the acceleration (instead of the position) is the index reduction.

Response: Rewritten quite a bit of the text. Added: "As a consequence of the index reduction, the accelerations of the point masses appear in the algebraic equations." These equations also still include position and velocity of the point masses. Therefore, I found the added sentence a more precise description than only mentioning constraining the acceleration.

- A consequence of using index reduction schemes is that small inaccuracies in the integration could lead to a situation where the velocities and positions are no longer consistent with the constraints. I am guessing that your solver has methods to correct for this? I don't know if it is possible, but I would be curious if your simulations had significant drift. If your dynamics give position and velocity, this can easily be checked with equations 13-16, you could make a statement of how well the position and velocity constraints were respected.

Response: The largest tether/bridle lenght drift at the end of the simulation is 1e-4 m. The consistency conditions are satisfied with the choice for the initial state and are not actively corrected by the solver in the current implementation. Given the small drift, the algebraic eqs. are sufficiently effective in satisfying the consistency conditions throughout the simulation. Added: "The solver produces a consistent simulation with insignificant drift in consistency conditions, i.e., the distance between the wing and the KCU drifts with 0.0001~\unit{m} in 24.2~\unit{s}."

- Also, to help gauge the quality of the dynamic solution, I would like to know more about the integration scheme. Presumably, those software packages that you use give the theory. Is it an implicit vs. explicit? constant time step vs. variable time step? Runge-Kutta? Generalized Alpha? What order? Is there numerical dissipation?

Response: Rewritten paragraph: "The DAE is transformed into a semi-explicit form and solved with the IDAS integrator in CasADi~\citep{Andersson2018}. IDAS employs the backward differentiation formula (variable-order, variable-coefficient) for implicit integration to solve the system. The motion is resolved at a fixed time step of 0.1 seconds."

- Another problem I have is the fact that equation 12 doesn't actually describe the problem that you are solving... line 232, you explain that you use prescribed accelerations from one of the point masses. I think it's ok as a way of tryin to back out further dynamic information from the acceleration measurements ... but I think it would be better to have a diagram, showing that set-up exactly (i.e. with a boundary condition triangle) and show the equivalent equation 12 for that set-up. When you prescribe acceleration for one of the masses, then you in effect seperate the problem into two independent calculations: 1) A kinematic calculation, based on acceleration to get the wing position and velocity 2) a dynamic calculation for the second point mass.

Response: Good point. Added the "the equivalent equation 12". I'm not sure what you mean with boundary condition triangle. However, I expect that with the rewritten DAE (the one that is actually solved) the method is described precisely.

- Equation 12 shows dependency on the position and the velocity. However, it's not clear to me what values were used for this calculation. Given the acceleration solution from 12, one could integrate to get the velocity and position. If this is the case, further description of the integration scheme would be needed. Another alternative is the kinematic data that you describe in section 2 and appendix A. Was this used in place of integrated quantities? If this is the case, then my earlier point about constraint drift might not be applicable. This detail is also important to me because it's not clear to me whether position and velocity is an input or an output of this dynamic model.

Response: The first suggested option was used. Added introduction of the differential states, algebraic states, and control inputs at the start of the section and the time derivative of the state vector towards the end of the section.

- What is important is differentiating between what is prescribed and what is solved in the dynamic model and this is very unclear by this description.

Response: should be better now that the variables are made explicit.

- In the results section, it appears that you use the dynamic model with 30 sections, not 1, I think it would be more appropriate to show equations and describe this model with N segments.

Response: a generic formulation with N segments would in our opinion be very abstract and not as illustrative. However, we acknowledge that it may be more valueable to present a model with more than 1 tether element. Therefore, now the equations of a 2-tether-element model are presented. For the generic model, the readers may consult the work of Zanon.

- Overall, I think the description of the dynamic model is a bit muddled and it is difficult to assess how results from this model could be used to understand and compared with the other model

Response: This is subtantially improved due to all modificiations stated above.

In both models, it's not clear whether tether force or tether elasticity is acounted for. Does tether force factor in any of the models? Does tether stretching factor into any of the models? If so, I am guessing you used standard linear relations? For the sake of reproducibility, what stiffness constants? In the quasi-static model, it would impact the sag solution. In the dynamic model, the net effect of tension and curvature is a lateral force. Equation 12 does not appear to show tension effects ... these effects could be important in the modelling, but there are no statements on these points. I also have similar concerns with the bridle, is there any deformation? in reality or in the model?

Response: The results in the paper don't include stretching. In a preliminary anaylsis, streching was assessed with the 'quasi-static' model. Because a thick, stiff tether (modulus of 48 GPa) is used for the size of the kite, the stretching is in the order of 0.1%. Added sentence to sec. 3.1: "The tether and bridle are assumed to be rigid and variations in length due to elastic effects are neglected."

Figure 9, consider using different dash pattern (long vs. short), to make the differentiation between the dynamic and the 1 element model more clear. Please describe in more precise terms the

directions, the terms 'Up-apparent wind' and 'Cross Apparent wind' is vague. You already have this wind reference frame ... when I see 'apparent wind', I guess that is a new reference frame, aligned with the wind relative to some body. What body? Does this reference frame vary along the tether? Or is it the apparent wind at one location, then the deflections are measured against that ... it's really unclear to how to interpret this geometry.

Response: Modified line style as suggested. Relative to the wing; modified caption: "Tether-kite lines with cross-axial displacement decomposed with respect to the tangential apparent wind velocity of the wing (see Figure 4)."

Figure 10 is also a little confusing due to lack of details in previous sections. In the turns, there are large discrepencies between all models for pitch. Since it's not clear what points of the wing each of these measurements correspond to, I am confused as to whether these discrepencies are due to measurements at different locations on a deforming structure, or errors in the models.

Response: Rephrased a little: "The pitch and roll of the kite derived from the attitude of the bridle element (with respect to the tangential plane) along the figure of eight. The results of the steady-rotation-state and dynamic analyses are depicted alongside the pitch and roll inferred from attitude measurements of the two sensors mounted to the wing, which include local effects of wing deformation."

Line 286, this statement is vague in light of my confusion over how position and velocity factor into the equations.

Response: Should be clarified after all modifications.

Line 288, The author makes a statement about discrepancies in the tether reel-out speed. reel-out acceleration is an input, so I must assume that the dynamic model integrates this quantity. Yet, the dynamic model described in section 3 does not say whether it calculates the tether lenght and velocity and how these quantities are integrated. So it's difficult for me to interpret these statements.

Response: Should be clarified after all modifications.

Line 289. The term tether slack is not well defined. A slack of 0.3 sounds more like an elastic deformation (i.e. a change in length), where as I think of slack as more of a lateral displacement of the cable.

Response: Changed text: "The reconstruction yields a slightly adapted tether reel-out speed with respect to the measured speed and imposes a constant difference between the tether length and radial kite position in the simulation. In this paper, we refer to this difference as tether slack."

Figure 11 and the argumentation on line 291: So the argumentation explains that the tether force depends on 'slack' I am guessing that this slack variable effects the average over time tether forces? As such, since this 'slack' is tuned, then the overall agreement in figure 11 is due to your choice for slack ... so it begs the question how you chose slack? if you chose slack by other data, then figure 11 agreement is a sort of validation, otherwise, it's not. Now the spikes that occur in the turns ... I am not completely sure about this point because I had difficulties interpreting appendix A, but the procedure that was used to generate the flight path appears to have a smoothing effect on the flight path... So it's totally expected that a smooth flight path would produce a more smooth dynamic

response, hence the failure to capture these spikes in the tether force. It's also unclear how ignoring turbulence or other unsteady aerodynamic forces could impact these results

Response: I changed the chosen slack value to the mean value of the steady rotation analysis to make the choice for the tether slack less arbitrary: "The initial tether length of the simulation is chosen such that the tether slack is $0.28\,\unit{m}$, which is the mean value observed in the steady-rotation-state results." It's actually the 'smoothened' dynamic response that shows more spikes, not the measurements - this was a typo in the earlier manuscript. Also the steady-state calculation with the same tether length input as for the dynamic simulation shows similar tether force spikes. Text added: "These differences are however not specific to the dynamic model but are expected to be artefacts of the wing and tether acceleration control input." Going more into detail is ought to be out of the scope of the paper.

It would be interesting to include results from a quasi-static model that ignores this extra point mass for the KCU and treats the whole wing bridle as a single point mass under rigid body motion. Such a comparison would show what you gain in terms of additional information by adding this 1 extra piece of complexity. This research question is implied in the introduction so it would be good to show that as well.

Response: (1) What do you mean with point mass under rigid body motion? I assume you simply mean that instead of using 2 point masses to model the kite, a rigid body is used encompassing the kcu, bridle, and wing. I expect that this would not make much difference. The added DOF introduced by the rigid body is the yaw angle of the kite. For the present study, this angle does not add any value. Morover, it is convenient to only use point masses in the model as also the tether is modelled with lumped masses. (2) Which research question is implied? The abstract mentions the single-point kite model; this is however not to be confused with a rigid body. The conclusions pertaining to a single-point kite model are ought to be sufficiently supported with the literature study.

Line 390: Stating that the dynamic model solves the actual motion is a bit strong. In previous parts, you say that the dynamic model isn't more accurate. I guess what you are trying to say is that is simulates the transient effects.

Response: rephrased: "In contrast to the dynamic model, the steady-rotation-state model neglects transient effects."

So this paper is interesting in showing the dynamic effect of the control unit, so this is good.

:)

This paper also introduces two models to help investigate these dynamics. In the introduciton, the author implies that these models could be used to better simulate these important dynamics. The author does present a lot of validation data for these two models. However, this reviewer feels that there are several sources of errors that are not adequately discussed in this paper. The author had to condition the experimental data, it's not clear how this conditioning effected the analysis. In the description of the study, many important assumptions and/or modelling details are not specified, so it's unclear how these could have contributed to the errors. Later the author implies that some things were tuned (i.e. slack), yet it's not clear what that means and how it enters the model, so it's confusing on how to interpret comparisons as validation or not. Also on pitch, due to lack of details on the locations of sensors and the assumptions of how the wing bridle deformation, it also difficult

to understand the differences in pitch in turns. I think the the author needs to state clearly the dynamics/kinematics the model predicted well and the parts where the models did achieve agreement. Then give some ideas on how that could be improved. To better improve the models and better improve the understanding of the dynamics, the author should suggest future work that is needed. I also think the author should discuss a little the role of unsteady aerodynamics on the discrepancies that they see in the tether forces.

Response: When you talk about the sources of errors, do you mean those related to the dynamic model? Without the conditioning of the experimental data, the dynamic simulation would not be possible. As indicated in the appendix: we acknowledge that the flight trajectory reconstruction might not be strictly valid. Nevertheless, it serves the higher aim of this study. Most raised topics are already covered in the previous responses. The aerodynamics are deliberately left out of the equations as much as possible (at least those of the wing). The tether forces discrepancies are believed to have different origins as mentioned in a previous response. Added a paragraph to the conclusion with ideas for improving the analysis.

The beginning of Appendix A, figure A1, all quantities are labelled symbolically, but there is no description on what these symbols mean.

Response: descriptions of symbols added in text and caption.

Also what sensor data was used for this, you have 2 GPS and 2 altimeters? or something else? Again past comments about the lack of details about the test set-up makes this confusing.

Response: Revised paragraph about flight data: "For this flight test, two Pixhawk sensor units were mounted to the wing, one on each of the two struts adjacent to the symmetry plane of the kite (red and green cases in Fig.2). The units are each equipped with an IMU, GPS sensor, and barometer for recording position and attitude. Only conditioned position data was made available by Kitepower based on measurements of sensor~0, which have been processed using the default Kalman filter implementation of the Pixhawk. The utilised velocity measurements come from the same sensor. However, for an unknown reason, this sensor did not measure acceleration. Therefore, the acceleration measured with sensor~1 is utilised."

I think that it's strange that A2 is described as an optimal control problem. This appears to be a type of least $R^2$ statistical curve fitting problem ... I can understand how the appearance is similar to an optimal control problem, but in my opinion it's a bit misleading to call it a control problem. There is nothing being controlled, you are trying to fit synthetic data to multiple points of measured data.

---

## Author Response (AR2)

Dear **reviewer #1**,

Thank you for the second round of reviewing.

In Figure 1, we have now combined close-up photos of the kite and the ground station during flight operation in the exact configuration that was used for the instrumented flight test introduced in Section 2. The flying kite in Figure 1(a) now complements the kite in launch position in Figure 2 and the CAD geometry of the kite in Figure 3. The three configurations are decisively different. Compared to the CAD geometry, the aerodynamically loaded membrane wing shows substantial deformations, especially noticeable here in the billowing of the canopy, which inevitably leads to a contraction of the trailing edge geometry (which is not visible). We think that for the introduction, this is a good first figure, giving an overview of the entire system, now in a lot more visual detail than before, and also including the ground station. That, considering that many readers will not be familiar with the technology. Figure 2 then visually outlines the experimental setup and sensor positions, but with the kite on the ground, and unloaded. Figure 3 introduces the bridle line layout, geometrical dimensions, reference frames, and terminology.

Dear **reviewer #3**,

Thank you for the second round of reviewing. We were aware of the points you addressed, but still searching to strike the balance between the level of detail of the explanations and readability. Thank you for your helpful suggestions.

Overall assessment:

This is my second time reading this work. Overall the description of the study and the model has been greatly improved and I am much more satisfied reading it. So, I don't have further comments on this section of the paper. Concerning the aim of this paper, (understanding how the mass of the KCU effects the turning and the development of models to account for this), the authors have definitely made some important progress in this direction, so this is a good paper. However, there is one point that I am uncomfortable with, on line 471 within the conclusions the authors state that the model resolves pitch and allows AOA computation. However, the results show large discrepancies when comparing numerical pitch and measured pitch, so I think that authors cannot make this point in the conclusions without further context or qualification (see notes below). Furthermore, I still think that the discussion around the source of errors can be further improved with 1 or two sentences at different points in the text. I encourage a more broad discussion on the source of errors at specific points in the text, but at the same time I can see a broader discussion in other places of the paper. So these criticisms might be more opinionated in nature, so I leave it up to the authors to decide whether these suggestions would improve the paper.

Important concerns on the conclusions

Starting at line 471: The authors state that resolving pitch allows for accurate AOA. However, the comparisons given in the paper are unsatisfactory. So I feel that it cannot be said in the conclusions that models lead to improved predictions for AOA, I think some further qualification and context is needed for this. Like the fact that 2 dof model could in theory give better AOA results, by better resolving pitch, however in this study there are still large differences and further study is needed to understand the sources of these errors. I think the authors need to be more precise to avoid (unintentionally) misleading the reader.

Response: We added some qualification to two paragraphs of the conclusion for which we differentiated between accuracy of solving the pitching motion and adequacy for determining the angle of attack:
".. since the developed model aims for simplicity to increase computational efficiency, it does not incorporate all relevant mechanical effects, such as tether elasticity. In addition to solving the motion dynamically, it could be necessary to refine the configuration of the kite model in order to increase the accuracy of solving the pitching motion and explain the observed differences between the measured and computed pitch."
and later:
"Further study is needed to assess how refined the pitching motion needs to be solved to accurately calculate the angle of attack of the wing."

Additional comments on the discussion (i.e. my more opinionated comments)

Line 342: Some speculation is given on the source of differences in the tether force between models and measurements. The hypothesis given, is that they are due to errors in the reconstruction. Another could also be that sharp variations of the tether force are attenuated by deformation and

damping effects that are not considered in the model. As an outsider to this study, it's not my place to give my own speculation ... but as an overall, a question in this study is to what extent are errors caused by shortcomings in the modelling, problems in the measurements or analysis of data? I would be curious to hear the authors opinions on additional sources of error.

Response: We expanded the explanation to make it more complete. Text added: "The wing control input being a source of error is affirmed by coinciding, unexpected tether length results computed with the steady-rotation states. Consequently, also the tether acceleration control input will be a source of error since it is derived from these results. Errors introduced by model deficiencies, such as neglecting kite deformation and elasticity and damping of the tether, are expected to be overshadowed by relatively high errors due to the input."

Starting at Line 375: There is a discussion on the differences between the pitch measurements and the predicted pitch in the models. There are significant differences in the turns. One factor is that in the model, the KCU bridal and wing assembly are treated as a rigid body in the model, the pitch is the orientation of that rigid body. Where as the measurements are from point locations on a wing that is undergoing deformation (this point on deformation is clear by the measurements). So I see a couple alternative explanations for these large differences. 1) the measurements are not sufficient to get an average pitch estimate of the whole assembly due to local deformation effects 2) the modelling fidelity is insufficient to get accurate predictions (i.e. wing/bridal deformation is important). Again, the authors hypothesize that the differences are attributed to the reconstruction errors ... Again it's not my place to speculate on what's really happening, I am just a reviewer ... but the point is that in the absence of supporting data, it would be better for future study to have a broader discussion on the different sources of error and limitations of both the model and the experiment. If the reader wanted to repeat this study and collect improved results, is it really just better reconstruction that they should focus on? Personally I am skeptical on this point ... but even on that point, you could run a sensitivity study on the reconstruction results to assess that point (i.e. do your results change significantly with small variations of reconstruction). I am not suggesting that such a study is carried out for this paper ... It's just that I am seeing speculation that isn't backed up by data, without consideration of alternate sources of error and the fact that analysis that could be used to back-up these assertions missing. So I feel the explanation of these errors could be strengthened.

Response: To be more complete, we addressed the different sources of error and qualification statements: ".. Contrastingly, the two models exhibit systematic differences during the turns, with the differences in pitch being particularly substantial. This can be attributed to substantial transient effects during the turns, which are disregarded by the steady-rotation states. Although the dynamic result lies closer to the average measured pitch during the turns, it does not exhibit a similar peak. This discrepancy is expected to be caused by multiple sources of error. First, the actual wing motion that is causing the peak in pitch during the turns might not be accurately reconstructed in the flight trajectory reconstruction. This is affirmed by the large imposed modifications due to the relatively high uncertainty of the position measurement during the turns. Second, the pitch of the kite assembly calculated with the rigidly-linked two-point kite model is inadequate to describe the desired pitching behaviour of the kite during this highly dynamic manoeuvre. Consequently, a higher-fidelity model might be needed to obtain a suitable, higher-resolution pitching motion description. And third, assuming that the kite model is adequate, the available measurements could be insufficient to estimate the pitch of the kite assembly due to the measurement of local wing deformation."

and

".. Despite including transient effects, the dynamic model does not necessarily show a better agreement with measurements than the steady-rotation-state model. This suggests that improving the method for solving the motion may not be effective unless the configuration of the kite model itself is refined to compute pitching motion more accurately. However, a definite conclusion cannot be drawn because the uncertainty of the measurements might distort the view of the model validity."

To debate the rigid-kite assumption more, Fig. 14 is added showing the deformations of the kite together with a paragraph in the "Kite attitude validation" section.

Section 4.4 deals largely with the same pitch problems that I have already highlighted from section 4.3. From my reading of figure 14, I feel the greatest error in pitch occurs during the turning manoeuvres in real-out, yet in the reel-in I feel that there is better agreement. This leads me to think that there is something occurring in the reel-out turns that is not adequately captured. I am not sure if these full cycles results can help explain this ...? If you agree I am curious if the authors could say something about this in the paper?

Response: we are not convinced that the pitch is more accurately solved during reel-in than during the straight sections in the reel-out phase. It is primarily the highly dynamic turns that give a poor agreement with the measurements. Therefore, we kept Sect. 4.4 as is.

Minor errors

Line 355: "The available measurements *are* useful ..."

Response: reformulated